## ARTICLES
## OPEN

# Prolonged activation of innate immune pathways by a polyvalent STING agonist

Suxin Li[1], Min Luo[1], Zhaohui Wang[1], Qiang Feng[1], Jonathan Wilhelm [1], Xu Wang[1], Wei Li[1], Jian Wang[1], Agnieszka Cholka[1], Yang-xin Fu[2], Baran D. Sumer [3], Hongtao Yu [1,4,5] and Jinming Gao [1,3 ✉]

The stimulator of interferon genes (STING) is an endoplasmic reticulum transmembrane protein that is a target of therapeutics for infectious diseases and cancer. However, early-phase clinical trials of small-molecule STING agonists have shown limited antitumour efficacy and dose-limiting toxicity. Here, we show that a polyvalent STING agonist—a pH-sensitive polymer bearing a seven-membered ring with a tertiary amine (PC7A)—activates innate-immunity pathways through the polymer-induced formation of STING–PC7A condensates. In contrast to the natural STING ligand 2′,3′-cyclic-GMP-AMP (cGAMP), PC7A stimulates the prolonged production of pro-inflammatory cytokines by binding to a non-competitive STING surface site that is distinct from the cGAMP binding pocket. PC7A induces antitumour responses that are dependent on STING expression and CD8[+] T-cell activity, and the combination of PC7A and cGAMP led to synergistic therapeutic outcomes (including the activation of cGAMP-resistant STING variants) in mice bearing subcutaneous tumours and in resected human tumours and lymph nodes. The activation of the STING pathway through polymer-induced STING condensation may offer new therapeutic opportunities.

STING plays a central role in innate immunity against infection and cancer[1–4]. STING is endogenously activated by cGAMP, a cyclic dinucleotide that is synthesized by cGAMP synthase (cGAS) in response to cytosolic DNA as a danger signal[5,6]. Activation of STING mediates a multifaceted type-I interferon (IFN-I) response that promotes the maturation and migration of dendritic cells (DCs), and primes cytotoxic T lymphocytes and natural killer (NK) cells for spontaneous immune responses[7–11]. In recent years, STING has emerged as an important target that activates antitumour immune pathways for cancer immunotherapy[12–17]. Previous studies have observed punctate structures after the addition of cGAMP to STING, indicating that oligomerization or even higher-order architecture may be critical for activation[18–21]. Therapeutic attempts to deliver cGAMP into the cytosol of target cells in which STING is located have been limited by its inherent properties as a small, dual negatively charged molecule[22]. Moreover, the rapid enzymatic degradation and clearance as well as off-target toxicity of cGAMP have hindered its further clinical application[23,24]. Thus, the pharmaceutical industry has devoted great efforts to the chemical modification of natural cyclic dinucleotides (CDNs) as well as developing new STING agonists to improve their bioavailability and pharmacological activity[25,26]. Despite therapeutic promise, several small-molecule agonists of STING have shown limited antitumour efficacy and dose-limiting toxicity in early-phase clinical trials[27,28].

Polyvalent phase condensation has been shown to regulate diverse biological processes, including ribosome assembly, gene expression and signal transduction[29,30]. Phase separation involves the assembly of macromolecular complexes through multivalent interactions[31]. A previous study has shown that DNA-induced liquid phase separation of cGAS triggers innate immunity[32]. By forming such biomolecular condensates, proteins involved in signalling cascades can be easily enriched in membraneless assemblies and amplify responses to small changes in the microenvironment. These biomolecular condensates are typically hundreds of nanometres to micrometres in size, and are transient and dynamic in response to specific stimuli or stress[33,34].

We previously synthesized a library of pH-sensitive polymers with linear or cyclic tertiary amine structures, among which a polymer with a cyclic seven-membered ring (PC7A) has shown a strong vaccine adjuvant effect through the STING-dependent pathway[17]. In this Article, we report that PC7A is a polyvalent STING agonist that acts through polymer-induced phase condensation of STING to activate an innate immune response with prolonged cytokine expression compared with cGAMP. The level of STING activation depends on the length of the polymer and thereby the valency of the interaction. We also demonstrate that PC7A nanoparticles (NPs) loaded with cGAMP lead to robust tumour growth inhibition and enhanced survival in two animal tumour models, and synergistic STING activation in resected human tumours and lymph nodes. We provide a proof of principle for new cancer immunotherapy strategies targeting the STING pathway.

## Results

**PC7A polymer activates STING with a spatiotemporal profile that is distinct from cGAMP.** To understand how PC7A-induced STING activation differs from cGAMP[20,21], we first investigated the intracellular distribution of GFP-labelled STING and the downstream signals in live cells in response to treatment. Remarkably, the temporal profile of PC7A-induced STING-puncta formation and maturation is distinct from those induced by cGAMP. When primed by cGAMP, STING-puncta formation occurs rapidly, producing a strong immune response that peaks around 6 h after stimulation, followed by rapid degradation and subsequent immune silence

[1]Department of Pharmacology, Harold C. Simmons Comprehensive Cancer Center, University of Texas Southwestern Medical Center, Dallas, TX, USA. [2]Department of Pathology, Harold C. Simmons Comprehensive Cancer Center, University of Texas Southwestern Medical Center, Dallas, TX, USA. [3]Department of Otolaryngology, Harold C. Simmons Comprehensive Cancer Center, University of Texas Southwestern Medical Center, Dallas, TX, USA. [4]Howard Hughes Medical Institute, University of Texas Southwestern Medical Center, Dallas, TX, USA. [5]Present address: Zhejiang Provincial Laboratory of Life Sciences and Biomedicine, School of Life Sciences, Westlake University, Hangzhou, China. ✉e-mail: Jinming.Gao@UTSouthwestern.edu

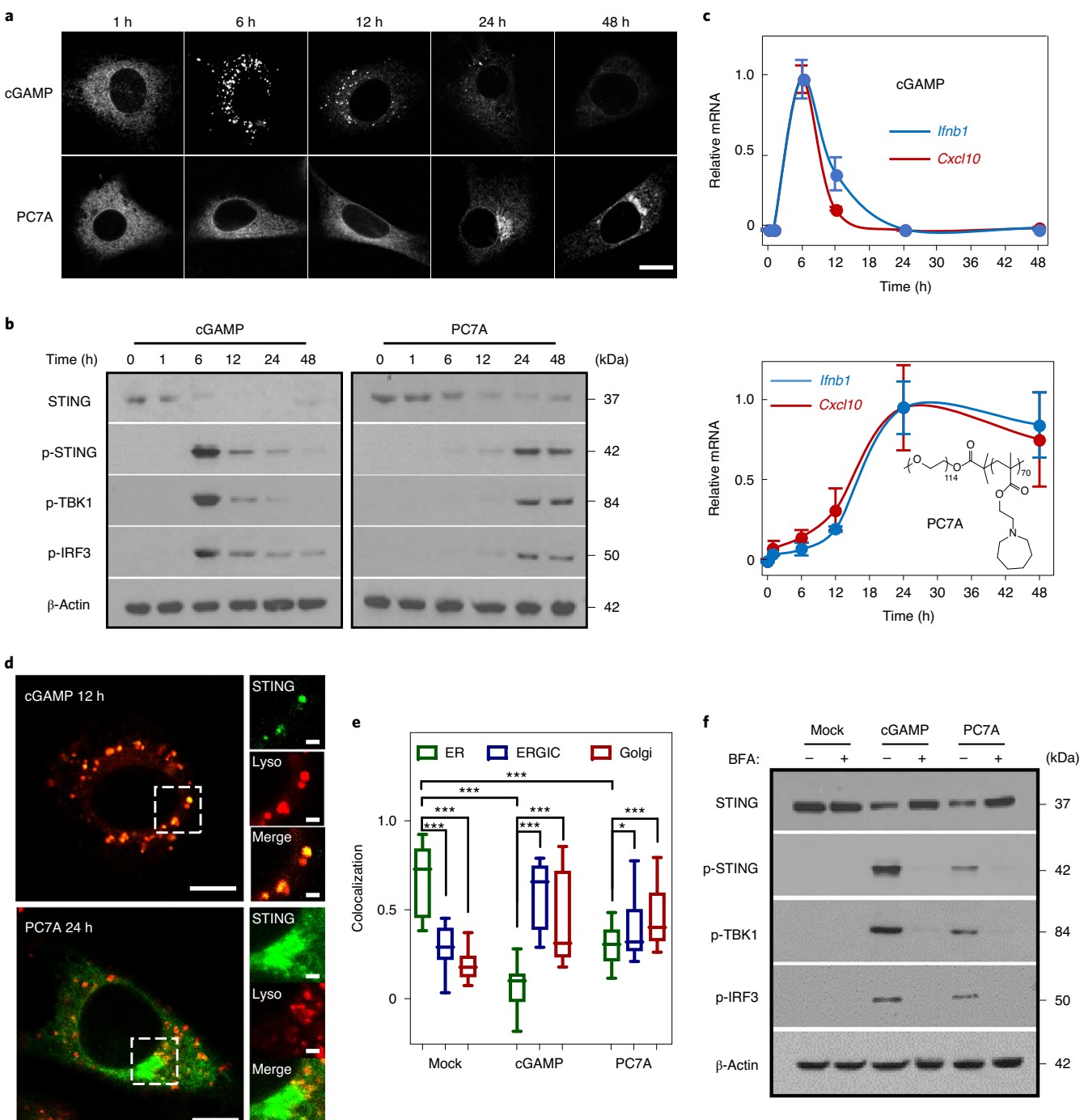

**Fig. 1 | PC7A polymer activates STING with a spatiotemporal profile that is distinct from cGAMP. a**, MEFs primed by cGAMP or PC7A exhibit different geometric and temporal patterns of GFP–STING-punctate formation and depletion. Cells were first incubated with cGAMP (10 μM, PEI was used for cytosolic delivery; Supplementary Fig. 1a,b) or PC7A micelles (10 μM) for 1 h, and the medium was then exchanged and cells were incubated for the indicated periods before imaging. Scale bar, 10 μm. **b**, THP1 cells that were treated with cGAMP display a burst effect of TBK1/IRF3 phosphorylation followed by rapid STING degradation, whereas treatment with PC7A led to sustained TBK1/IRF3 phosphorylation and slower STING degradation. **c**, Relative *Ifnb1* and *Cxcl10* mRNA levels show slower but prolonged STING activation in THP1 cells by PC7A compared with cGAMP. Data are mean ± s.d. *n* = 3 biologically independent experiments. **d**, GFP–STING colocalizes with lysosomes in MEFs 12 h after cGAMP treatment, supporting rapid degradation. By contrast, PC7A inhibits lysosomal degradation of GFP–STING, as indicated by the lack of colocalization and persistent GFP fluorescence. Scale bars, 5 μm (left images); 1 μm (right images). **e**, cGAMP and PC7A induce similar STING translocation from the ER to the ERGIC and Golgi apparatus. Colocalization was quantified using the Pearson's correlation coefficient. For the box plot, the centre line is the mean, the box limits show the 25th to 75th percentile, and the whiskers show the minimum and maximum values. *n* = 20 cells examined over three independent experiments. Statistical analysis was performed using two-tailed Student's *t*-tests (PC7A treatment group: ER versus ERGIC, *\*P* = 0.029; ER versus Golgi, \*\*\**P* = 0.0005; for all other comparisons, \*\*\**P* < 0.0001). **f**, STING translocation is necessary for downstream signalling as BFA, which is an inhibitor of protein transport from ER to Golgi, prevents the phosphorylation of TBK1/IRF3 by cGAMP or PC7A. The confocal images in **a** and **d** are representative of at least three biologically independent experiments.

(Fig. 1a–c). By contrast, PC7A generates a durable STING activation profile, with sustained expression of interferon-stimulated genes (*Ifnb1* and *Cxcl10*) over 48 h. STING degradation is delayed after PC7A stimulation, as indicated by the limited fusion of STING-puncta with lysosomes even at 48 h (Fig. 1d and Supplementary Fig. 1c). We observed a similar effect of delayed STING degradation in cGAMP-treated cells that were preincubated with bafilomycin A1—a vacuolar H⁺ ATPase inhibitor that blocks lysosome acidification—and in cells treated with combined treatment of cGAMP and PC7A (Supplementary Fig. 1d,e). Overall, these data suggest that the endo-lysosomal pH-buffering ability of PC7A may be responsible for slow STING degradation[35].

Despite the differences in size and kinetics of puncta formation, intracellular STING foci resulting from cGAMP or PC7A treatment follow a similar course of translocation from the endoplasmic reticulum (ER) to the ER–Golgi intermediate compartments (ERGIC) and the Golgi apparatus (Fig. 1e and Supplementary Fig. 2a). During transportation, STING forms clusters and phosphorylates TANK-binding kinase 1 (TBK1) and interferon regulatory factor 3 (IRF3; Fig. 1f), which initiates the downstream production of IFN-I proteins. In the presence of brefeldin A (BFA), which blocks protein trafficking between ER and Golgi[36], both cGAMP and PC7A fail to trigger the production of phosphorylated TBK1 and IRF3 (p-TBK1/p-IRF3) production and proinflammatory cytokine expression (Fig. 1f and Supplementary Fig. 2b–d).

**PC7A binds to STING and forms biomolecular condensates.** To investigate the biophysical mechanism of PC7A-mediated STING clustering and activation, we first determined the binding affinity between PC7A and STING (in human, amino acids 137–379, C-terminal domain) using isothermal titration calorimetry (ITC). STING binds strongly to PC7A (apparent dissociation constant ($K_d$ = 72 nM) but weakly to other polymers with the same backbone, such as PEPA (apparent $K_d$ = 671 nM; Supplementary Fig. 3a–c). Notably, polymers with cyclic side chains exhibit higher affinity to STING than linear analogues, and the seven-membered-ring of PC7A elicits the strongest binding. To investigate whether PC7A was sufficient to induce clustering of STING in vitro, we incubated cyanine-5 (Cy5)-labelled STING C-terminal domain (CTD) dimer with PC7A or PEPA at pH 6.5 (both P7CA and PEPA have apparent p$K$a values at 6.9, and remain as cationic unimers at pH 6.5). PEPA was used as a negative control due to its poor binding affinity to STING. After mixing of Cy5–STING and PC7A, liquid droplets were observed within minutes and grew over time; approximately 85% of STING proteins were present in the condensates after 4 h (Fig. 2a). Incubation of Cy5–STING with PC7A labelled with aminomethylcoumarin acetate confirmed colocalization of PC7A with STING puncta (Fig. 2b). Similar condensates were also observed in GFP–STING-expressing cell lysates after PC7A incubation (Supplementary Fig. 3d). The biomolecular condensates are hydrophobic as indicated by the increased fluorescence intensity and red-shifted maximum emission wavelength in a Nile-Red assay[37] (Supplementary Fig. 3e). Fluorescence resonance energy transfer (FRET) from GFP–STING to tetramethylrhodamine (TMR)–PC7A further confirmed the formation of a biomolecular condensate consisting of PC7A and STING in mouse embryonic fibroblasts (MEFs) overexpressing human STING (Fig. 2c). The downstream protein product p-TBK1 was also found in this macromolecular cluster (Fig. 2d). By contrast, no obvious STING condensation or activation was observed when PEPA was used in these studies (Fig. 2 and Supplementary Fig. 3d). At pH 7.4, few PC7A–STING condensates were formed (Supplementary Fig. 3f) due to micellization of PC7A polymers above its p$K$a (6.9) and PEG shielding[38,39].

**PC7A induces STING activation through polyvalent interactions.** Recent studies revealed that STING oligomerization after

cGAMP binding is responsible for the recruitment and activation of downstream TBK1 and IRF3 proteins[18–21]. We hypothesized that PC7A polymer can serve as a supramolecular scaffold and directly engage polyvalent interactions to multimerize STING molecules for activation (Fig. 3a). To test this idea, we first labelled STING proteins using a FRET pair (TMR and Cy5) and mixed the two differentially labelled proteins at a ratio of 1:1. After addition of PC7A, we observed strong energy transfer from TMR to Cy5 (Supplementary Fig. 4a), indicating close proximity of STING dimers after polyvalent binding to PC7A. Fluorescence recovery after photobleaching (FRAP) experiments[40,41] on STING–PC7A condensates revealed that, although both PC7A polymer and STING protein are exchangeable with surrounding molecules, PC7A exhibited a slower recovery rate than STING (Fig. 3b and Supplementary Fig. 4b,c).

To examine the effects of binding valence, we synthesized a series of PC7A polymers with an increasing number of repeating units. PC7A(*n*) refers to a polymer with *n* repeating units of the C7A methacrylate monomer. We incubated PC7A of increasing lengths with STING dimer under a matrix of concentrations in vitro to generate a phase diagram, which shows a minimum requirement of 20 repeating units for condensation (Fig. 3c). No phase separation was observed for PC7A(10). A higher degree of PC7A polymerization resulted in larger condensates (Fig. 3d and Supplementary Fig. 5a–c). For PC7A(110), more than 90% of STING proteins were found in the condensates, compared with 17% when PC7A(20) was used (Supplementary Fig. 5d). PC7A with a higher degree of polymerization exhibited lower phase reversibility and slower recovery rate of STING after photobleaching (Supplementary Fig. 5e,f). To investigate the relationship between condensate formation and STING activation in live cells, we treated THP1 cells with PC7A of varying lengths, and compared the *Cxcl10* mRNA expression levels. Longer polymers induced higher *Cxcl10* expression, with peak levels observed at 70 repeating units of PC7A (Fig. 3e). Further elongation of chain length (for example, 110) led to reduced *Cxcl10* expression, probably owing to the weaker signalling capacity of oversized condensates with excessive cross-linking and poor molecular dynamics[41,42].

**PC7A binds to a distinct surface site from the cGAMP-binding pocket.** The STING–PC7A condensates are sensitive to high concentrations of salt or the presence of other proteins. Although STING–PC7A condensates were formed at a physiological concentration of NaCl (150 mM), no phase separation was observed when the salt concentration was raised to 600 mM (Supplementary Fig. 6a,b). When bovine serum albumin (BSA) was added, the condensates decreased in number and size (Supplementary Fig. 6c). To further investigate the specificity of PC7A-induced condensate, we labelled STING with Cy5 and BSA with boron-dipyrromethene (BODIPY) dyes. In the presence of PC7A, only Cy5–STING was present in the condensates, whereas the majority of BODIPY–BSA was excluded (Supplementary Fig. 6d). As controls, mixtures of BSA and PC7A, or STING and BSA did not form condensates.

On the basis of the pH (Supplementary Fig. 3f) and salt (Supplementary Fig. 6a,b) effects on the PC7A–STING interactions and computational modelling (data not shown), we hypothesized that negatively charged surface sites on STING may be responsible for PC7A binding. To test this hypothesis, we constructed STING mutants with several negatively charged amino acids in the α5–β5–α6 region replaced by alanine and investigated their PC7A-binding affinity, phase condensation and STING activation both in vitro and in live cells. Notably, the mutation of two acidic residues (E296A/D297A) on the α5 helix was sufficient to abolish polymer binding and biomolecular condensation, whereas two other mutants (D319A/D320A and E336A/E337A/E339A/E340A) exhibited marginal effects (Fig. 4a,b and Supplementary Table 1). We next transfected HEK293T cells with mutant STING plasmids

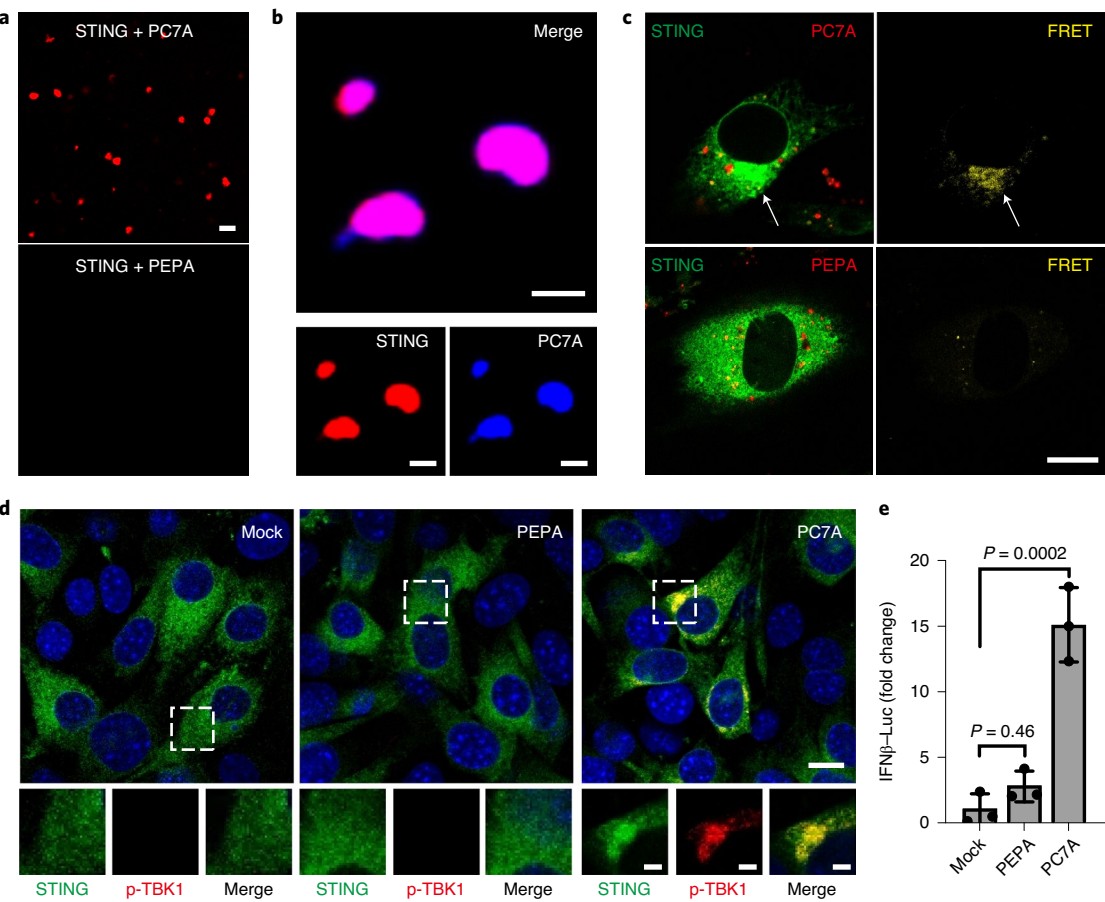

**Fig. 2 | PC7A polymer induces STING condensation and immune activation. a**, PC7A, but not PEPA, induces STING (Cy5-labelled) phase condensation after 4 h incubation. Scale bar, 10 μm. **b**, STING (4 μM, Cy5-labelled) and PC7A polymer (2 μM, aminomethylcoumarin acetate-labelled) are colocalized within the condensates. Scale bars, 5 μm. **c**, Hetero-FRET from GFP–STING to TMR–PC7A shows the colocalization of STING and PC7A in MEFs. Energy transfer was not observed from GFP–STING to TMR–PEPA. Cell culture conditions were identical to those described in Fig. 1. GFP ($\lambda_{ex}/\lambda_{em} = 488$ nm/515 nm) and TMR ($\lambda_{ex}/\lambda_{em} = 555$ nm/580 nm) signals are shown on the left in green and red, respectively. FRET signals ($\lambda_{ex}/\lambda_{em} = 488$ nm/580 nm) are shown in yellow on the right. Scale bar, 10 μm. **d**, p-TBK1 is recruited into the STING–PC7A condensates. Scale bar, 10 μm. Insets: magnified images of the areas indicated by the white boxes. Scale bars, 2 μm. **e**, PC7A, but not PEPA, induces the expression of IFNβ–luciferase in ISG-THP1 cells. Data are mean ± s.d. $n = 3$ biologically independent experiments. Statistical analysis was performed using one-way analysis of variance (ANOVA). The confocal images in **a–d** are representative of at least three biologically independent experiments.

and measured downstream activation. Consistent with the abrogation of PC7A binding and condensation, the E296A/D297A mutant was deficient in forming condensate structures and inducing TBK1 phosphorylation and *Ifnb1/Cxcl10* expression in cells (Fig. 4c and Supplementary Fig. 7a,d). By contrast, these STING mutants did not impact cGAMP-mediated STING activation (Supplementary Fig. 7b,c). Together, these data suggest that the Glu 296-Asp 297 site on the α5 helix of STING, which is distinct from the cGAMP-binding site, is responsible for PC7A binding and induced activation.

Endogenous STING agonists (cGAMP or other CDNs) bind to the STING dimer interface covered by a lip of four-stranded antiparallel β-sheet (human amino acids 219–249)[43,44]. A natural STING variant (R232H) that occurs in ~14% of the human population exhibits a reduced response to small-molecule STING agonists[45]. As PC7A binds to a STING site that is different from the cGAMP-binding pocket, we tested the biological activity of PC7A in THP1 cells harbouring the STING R232H variant. Whereas the cGAMP response was abrogated in these cells as expected, PC7A still had the ability to elevate IFNβ–luciferase expression (Fig. 4d). Additional studies in mutant HeLa cells (R238A/Y240A or Q273A/A277Q mutations that abolish cGAMP binding or prevent STING oligomerization after cGAMP binding, respectively)[20,21] show persistent

PC7A-induced STING activation, whereas cGAMP-mediated effects were abolished (Fig. 4e,f and Supplementary Fig. 7e–h). Collectively, these results demonstrate that PC7A stimulates STING through cGAMP-independent mechanisms.

**PC7A prolongs innate activation in vivo and synergizes with cGAMP in antitumour immunity.** In vitro cell culture studies show that PC7A NPs generated more-durable STING activation compared with free cGAMP (Fig. 1a–c). To test whether PC7A NPs prolong STING activation in vivo, we intratumourally injected cGAMP (50 μg), PC7A NPs (50 μg) and cGAMP-loaded PC7A NPs (2.5 μg/50 μg) into MC38 tumours (~100 mm³) and measured the expression of interferon-stimulated genes in both tumours and the draining lymph nodes over time. Owing to the ability of PC7A for STING activation and cytosolic delivery of cGAMP, we chose a lower dose of cGAMP (~5 wt% loading) in cGAMP–PC7A NPs for the majority of in vivo studies. cGAMP–PC7A NPs were prepared using a base titration method, resulting in spherical micelles of 29.9 ± 2.5 nm (mean ± s.d.) in diameter and over 90% loading efficiency (Supplementary Fig. 8). Consistent with our in vitro studies, mice treated with free cGAMP showed rapid *Ifnb1/Cxcl10* expression 6 h after intratumoural injection, while the activity

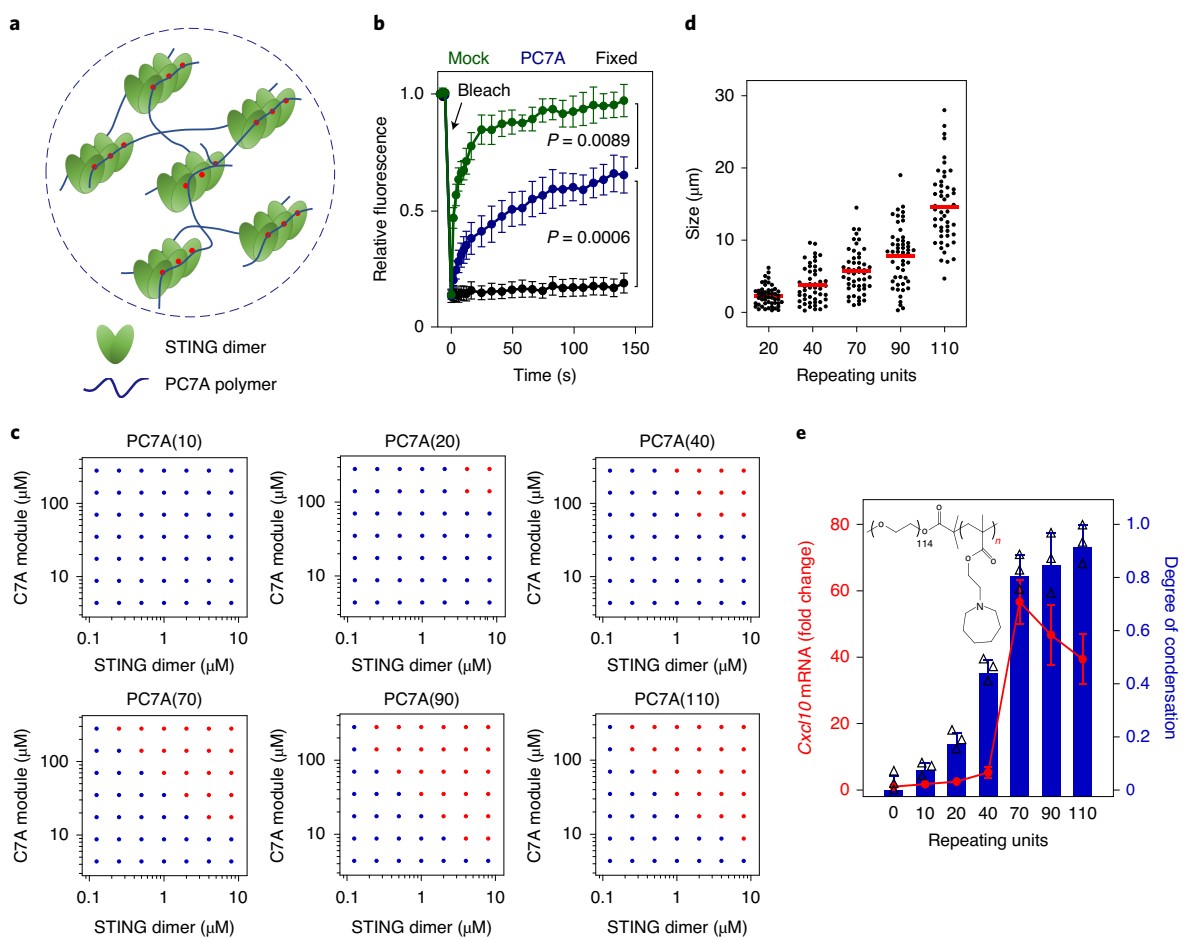

**Fig. 3 | PC7A polymer induces STING condensation and immune activation through polyvalent interactions. a**, Schematic of STING oligomerization and condensation driven by PC7A through polyvalent interactions. **b**, PC7A decreases the molecular mobility of GFP–STING in the condensates compared with free GFP–STING in MEFs. Bleaching was performed 24 h after PC7A treatment, and recovery was monitored over 150 s. Untreated (mock) and fixed cells were used as mobile and stationary STING controls, respectively. Data are mean ± s.d. $n = 5$ cells examined over two independent experiments. Statistical analysis was performed using one-way ANOVA. **c**, Biomolecular condensation of STING and PC7A is dependent on PC7A valency. The red dots indicate phase separation, and the blue dots indicate no phase separation. **d**, Size distributions of STING condensates induced by an increase in PC7A with higher PC7A valency. Condensate size was calculated as the average of longest and shortest axes. $n = 50$ condensates examined over two independent experiments. The red lines represent the average. **e**, STING activation in THP1 cells is correlated with the PC7A valency, with optimal *Cxcl10* expression induced by PC7A(70). Data are mean ± s.d. $n = 3$ biologically independent experiments. For the experiments shown in **c–e**, polymers with different repeating units were used at the same C7A modular concentrations.

decreased considerably over 48 h in both tumour and nodal tissues (Supplementary Fig. 9). By contrast, PC7A-induced STING activity was minimal at 6 h but reached the maximum level at 24 h. cGAMP–PC7A NPs yield the most optimal STING activity profile, which exhibited a rapid increase in *Ifnb1*/*Cxcl10* expression compared with PC7A (50 μg) at 6 h and, in contrast to free cGAMP, this response was also sustained over 48 h.

Next, we investigated the antitumour efficacy in MC38 and TC-1 tumour models (Fig. 5). In MC38 tumours, we performed three intratumoural injections of free cGAMP (2.5 μg or 50 μg; high-dose data are provided in Supplementary Fig. 10a), PC7A NPs (50 μg) or cGAMP–PC7A NPs (2.5 μg or 50 μg) when tumours reached ~50 mm³ in size. As a negative control, we injected mice with a 5% glucose solution (all of the treatment groups were prepared in 5% glucose solutions). The results show that all of the mice in the control group died within 50 d after MC38 inoculation. Groups that were treated with cGAMP (2.5 μg) or PC7A alone showed notably extended the survival compared with the control group, while the difference between the two treatment groups was not statistically significant. cGAMP–PC7A NP treatment achieved the most

efficacious outcome, with 4 out of 7 mice remaining tumour free over 100 d after tumour inoculation. In the TC-1 model, all of the mice in the control group died within 26 d. cGAMP or PC7A alone conferred a minor degree of immune protection, extending the median survival time by 4 d or 8 d, respectively. The cGAMP–PC7A NP treatment showed significantly improved tumour growth inhibition and long-term survival compared with either treatment alone.

In MC38 tumours, a high dose (50 μg) of free cGAMP treatment did not lead to significantly improved tumour growth inhibition compared with the low-dose group (2.5 μg; Supplementary Fig. 10a). By contrast, systemic side effects were observed at the higher cGAMP dose, as evidenced by the elevated levels of alanine transaminase and aspartate transaminase (liver), urea (kidney) and systemic cytokine (for example, IL-10; Supplementary Fig. 10b–e). cGAMP–PC7A NP treatment did not show a significant increase in toxic side effects compared with the control group.

Previous studies have shown an association between elevated IFN-I production and increased tumour infiltration of PD-1+ cytotoxic T lymphocytes[7,46–48]. We hypothesized that STING activation by cGAMP–PC7A NPs may synergize with PD-1 blockade.

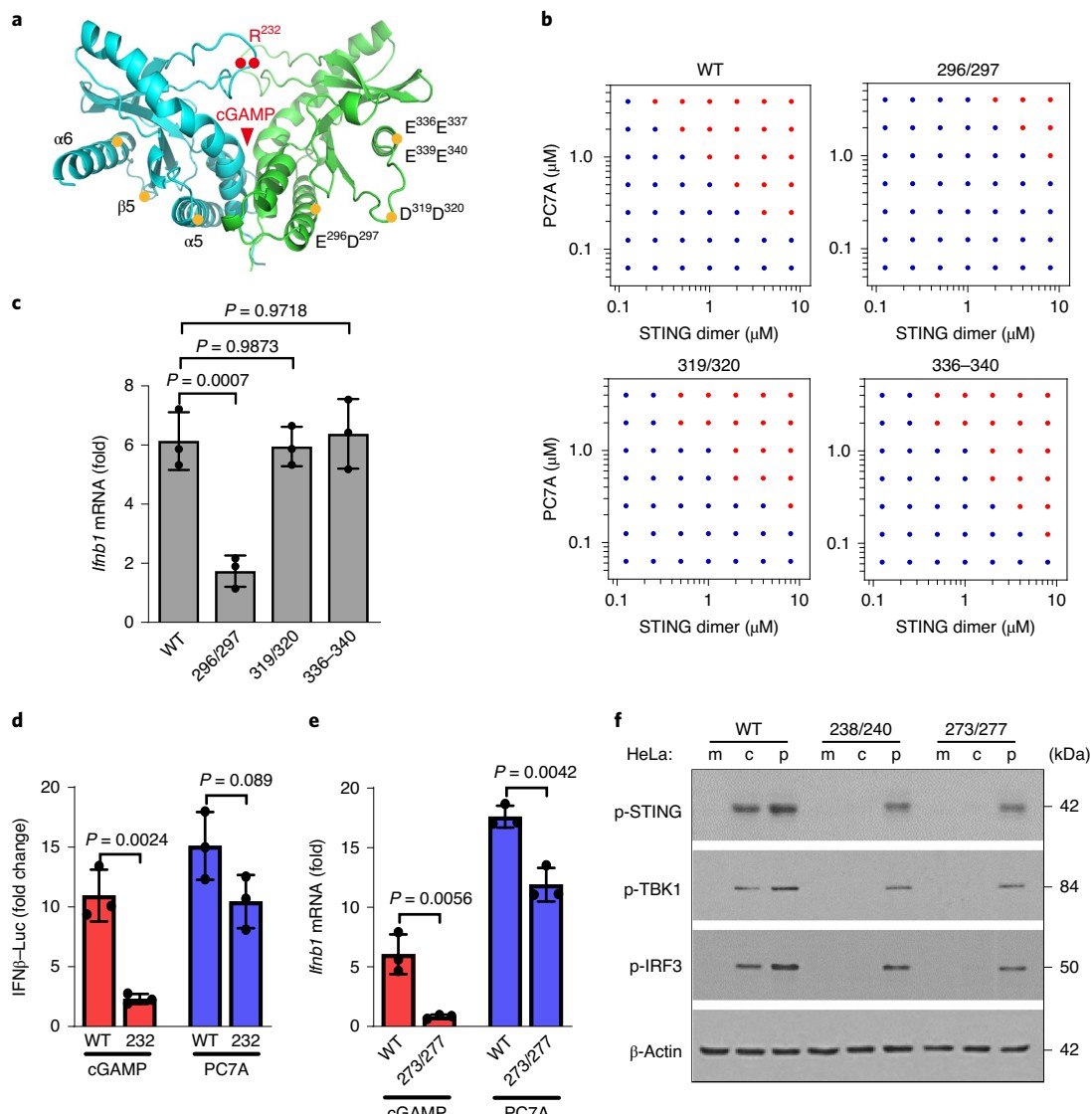

**Fig. 4 | STING condensation and activation by PC7A polymer occurs through a distinct binding site from cGAMP. a**, Schematic of site-directed mutagenesis on the STING structure. Mutation sites are distinct from the cGAMP-binding pocket. **b,c**, STING mutation of E296A/D297A abolishes STING condensation (**b**) and immune activation (**c**) in response to PC7A. Other mutations in STING do not affect PC7A-induced STING activation. Data are mean ± s.d. *n* = 3 biologically independent experiments. Statistical analysis was performed using one-way ANOVA. **d–f**, PC7A retains immune activity in several cGAMP-resistant STING variants. R232H in THP1 cells (**d**) or R238A/Y240A in HeLa cells (**f**) abrogates cGAMP binding. Q273A/A277Q (**e,f**), which disrupts the tetramer interface and cGAMP-mediated STING oligomerization, abolishes STING activation by cGAMP but not by PC7A. Data are mean ± s.d. *n* = 3 biologically independent experiments. Statistical analysis was performed using two-tailed Student's *t*-tests. c, cGAMP; m, mock; p, PC7A polymer.

We found that the combination provided significantly improved efficacy—100% of the mice remained tumour-free after 100 d in the mouse MC38 colorectal tumour model (Supplementary Fig. 11a–c). The therapeutic efficacy was also improved in the more aggressive TC-1 tumour model; more than 50% of the mice bearing TC-1 tumours survived for longer than 45 d (Supplementary Fig. 11d–f).

**STING status and immune cell type on PC7A-induced antitumour immunity.** Using an in vivo cell killing assay, our previous study showed that the generation of antigen-specific T cells by the PC7A NP vaccine was dependent on the STING–IFN-I pathway[17]. To confirm the importance of the STING pathway and to determine whether host or cancer cell STING status has a more dominant role in PC7A-induced antitumour immunity, we performed a tumour

growth inhibition assay in host *Tmem173*[−/−] (which encodes STING) mice + wild-type (WT) MC38 tumours and WT mice + *Tmem173*[−/−] MC38 tumours (Supplementary Fig. 12a–c). Without treatment, WT MC38 cancer cells grew faster in *Tmem173*[−/−] mice than in WT mice, indicating the role of the STING pathway in immune protection by the host alone. The antitumour efficacy improvement of PC7A and cGAMP–PC7A NPs was abolished in *Tmem173*[−/−] animals compared with the WT mice. By contrast, comparable antitumour efficacy by PC7A and cGAMP–PC7A NPs was observed when treating WT mice with *Tmem173*[−/−] tumours versus WT MC38 tumours.

To further investigate the immune-cell-dependent contribution to antitumour immunity, we evaluated tumour growth inhibition by antibody blockade of CD8 T cells, NK cells and in CD11c-DTR

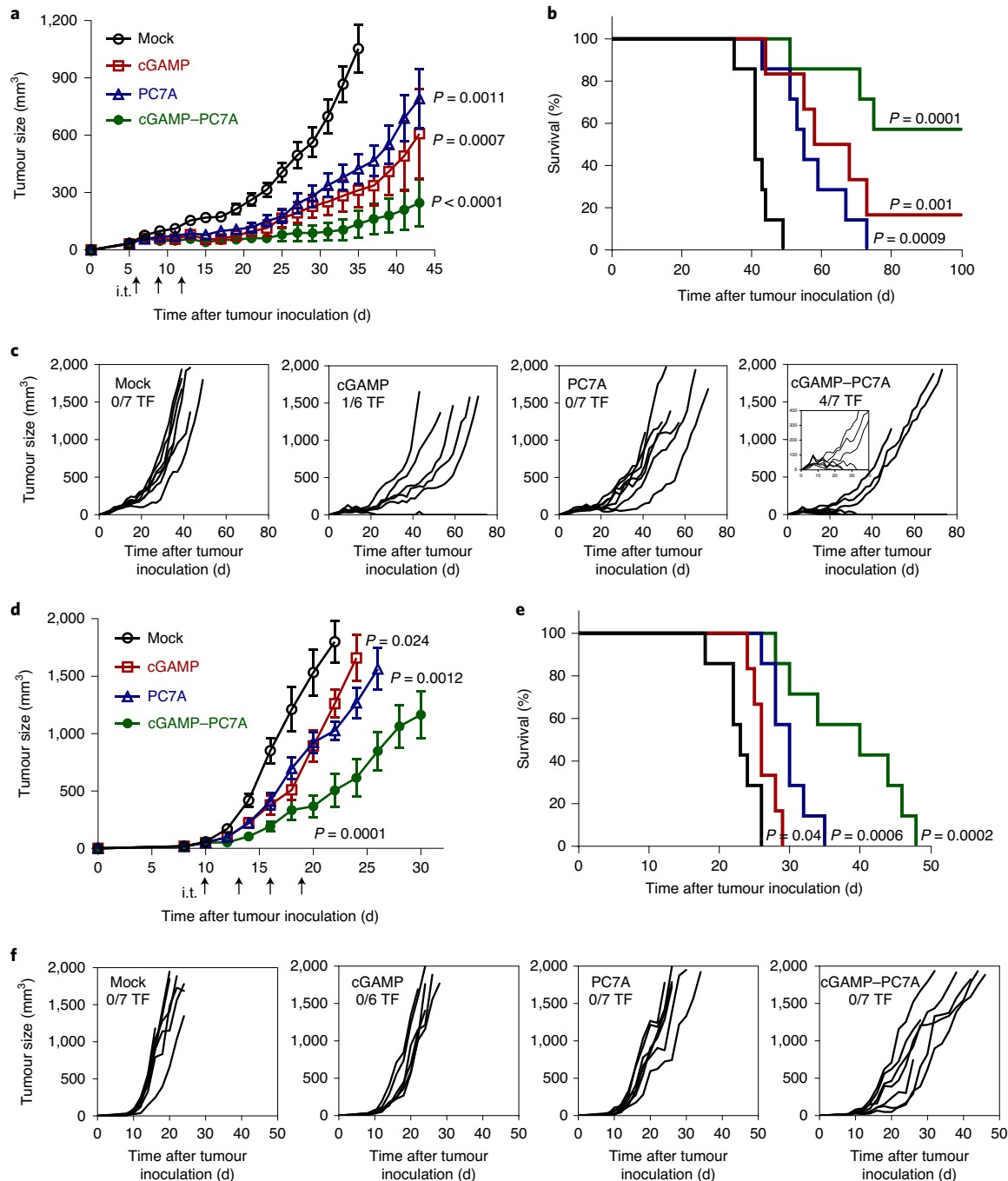

**Fig. 5 | PC7A and cGAMP show synergistic antitumour efficacy in tumour-bearing mice. a–f**, MC38 (**a–c**) and TC-1 (**d–f**) tumour-bearing mice were injected intratumourally (i.t.; arrows) with 5% glucose (mock), cGAMP (2.5 μg), PC7A NPs (50 μg) or cGAMP-loaded PC7A NPs at the indicated time points. Mean tumour volume (**a,d**), Kaplan–Meier survival curves (**b,e**) and spider plots of individual tumour growth curves (**c,f**) are shown. PC7A NPs or cGAMP alone offer some degree of immune protection. cGAMP-loaded PC7A NPs confer a synergistic antitumour immune response, with significantly improved survival; 4 out of 7 mice in the MC38 model were tumour-free (TF). In the tumour growth studies (**a** and **d**), data are mean ± s.e.m. $n = 7$ (mock), $n = 6$ (cGAMP), $n = 7$ (PC7A) and $n = 7$ (cGAMP–PC7A) biologically independent mice in each tumour model. For **a** and **d**, statistical analysis was performed using two-tailed Student's $t$-tests (versus mock). For **b** and **e**, statistical analysis was performed using Mantel–Cox tests.

transgenic mice with depletion of DCs[49]. Blockade of CD8 T cells abolished the antitumour efficacy of PC7A treatment whereas blockade of NK cells showed minimal effect (Supplementary Fig. 12d,e). Results in CD11c-DTR mice showed that DC depletion reduced the therapeutic efficacy after treatments, albeit to a lesser extent when compared with CD8 T-cell blockade (Supplementary Fig. 12f).

**STING activation in human tissues.** To examine the translational potential, we investigated the feasibility of STING activation in human tissues. We acquired freshly resected squamous cell carcinoma from the base/lateral of tongue, cervical tumour tissues and a sentinel lymph node. We locally injected these tissues with cGAMP, PC7A NPs or cGAMP–PC7A NPs, incubated them in cell culture medium for 24 h at 37 °C, and detected IFN-related gene expression.

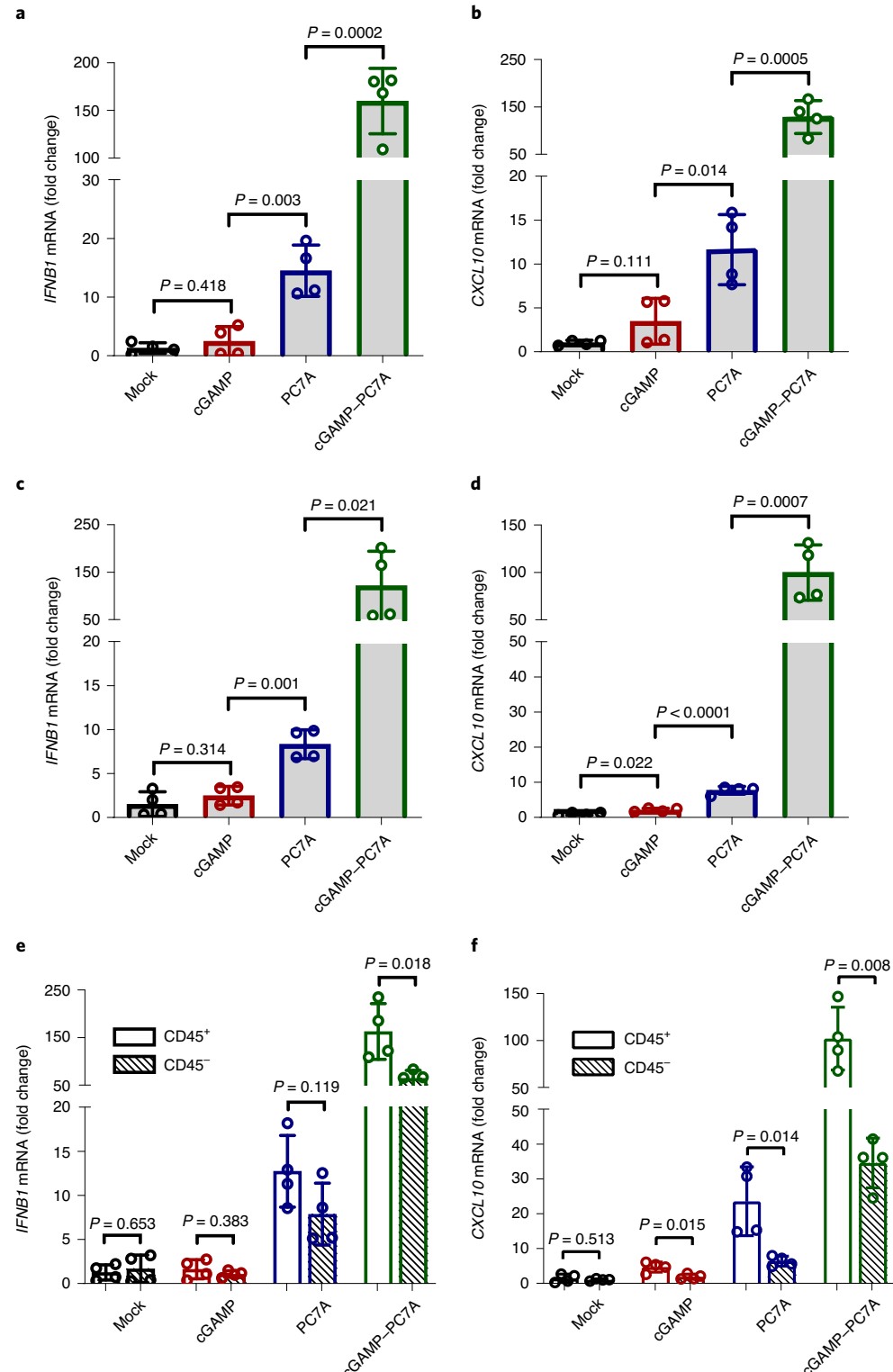

**Fig. 6 | PC7A and cGAMP show synergistic STING activation in fresh human tissues. a–f**, Free cGAMP alone is unable to activate STING, whereas PC7A NPs and cGAMP-loaded PC7A NPs demonstrate effective STING activation. Fresh surgically resected sentinel lymph nodes (SLN) (**a,b**) or squamous cell carcinoma samples from the base of tongue (SCC-BOT) (**c–f**) were divided into multiple sections (1–5 mm³) and injected with 5% glucose, free cGAMP, PC7A NPs or cGAMP-loaded PC7A NPs in 5% glucose solutions. *IFNB1* (**a,c,e**) and *CXCL10* (**b,d,f**) gene expression was measured after 24 h incubation. **e,f**, The CD45⁺ cell population exhibited an enhanced level of STING activation compared with CD45⁻ cells. Data are mean ± s.d. *n* = 4 SLN sections from the same patient in **a** and **b**. *n* = 4 SCC-BOT tumour sections from the same patient in **c–f**. Statistical analysis was performed using two-tailed Student's *t*-tests.

Free cGAMP had a marginal effect on *IFNB1* and *CXCL10* mRNA expression over the control due to limited bioavailability. By contrast, PC7A NPs elevated downstream signals by fivefold to twentyfold. A substantial increase in cytokine expression (100–200-fold; Fig. 6a–d and Supplementary Fig. 13) was observed after treatment with cGAMP–PC7A NPs in all of the tissue types. Notably, after treatment with PC7A NPs and cGAMP–PC7A NPs, CD45+ myeloid cell populations in the tumour showed a higher level of STING activation compared with CD45− cells (Fig. 6e,f), indicating that leukocytes, rather than cancer cells, are the primary targets for STING-mediated immunomodulation by NPs.

## Discussion

The therapeutic strategies described in this study take advantage of non-canonical STING activation by a synthetic polymer with cell intrinsic pathways by cGAMP for cancer immunotherapy. First, we determined a distinctive surface-binding site on the STING protein by the PC7A polymer that is different from the binding pocket of cGAMP or other CDNs. Although a previous report showed that the PC7A NP vaccine worked through STING for T-cell activation[17], it is not clear whether PC7A competes with cGAMP for the same binding site at the STING dimer interface. The discovery of non-competitive binding sites formulates a basis to combine PC7A with cGAMP for synergistic STING activation while enabling PC7A to activate cGAMP-resistant STING variants (Fig. 4d–f). In humans, STING consists of several haplotypes (for example, 14% of the human population have the R232H phenotype) that exhibit reduced innate activity in response to CDN agonists[50,51]. PC7A presents an alternative STING activation strategy in these STING-variant patient populations. Second, we uncovered a PC7A-induced protein condensation mechanism for STING activation. We used a synthetic polymer to induce polyvalent phase condensation for biological activation. Phase condensates are shown to impact a broad range of biological processes and are under intensive investigations in biophysics and cell biology[29,30]. Here we provide a proof of concept to install polymer-induced protein condensation as an emerging bioengineering principle for biological interrogation and pharmaceutical development.

STING remains a promising target for cancer immunotherapy, but several small-molecule STING agonists showed limited efficacy and dose-limiting toxicity in early-stage clinical trials[27,28]. Here, intratumoural injection of a high dose of cGAMP (50 μg) did not lead to significant tumour growth inhibition compared with a low dose (2.5 μg), but resulted in increased systemic toxicity (Supplementary Fig. 10), corroborating clinical observations. We attribute the limited therapeutic window to the poor pharmacokinetics and mechanism of action in STING activation. Owing to its small size (674 Da) and water solubility, blood perfusion can quickly remove cGAMP from the tumour site to systemic circulation, limiting STING activation to a few hours inside tumours (Supplementary Fig. 9).

Compared with cGAMP, the PC7A polymer induces a slower but more sustained STING activity in vitro and in vivo. We attribute this kinetic difference to several factors. First, endosomal escape followed by cytosolic transport to reach ER-bound STING target is probably faster for cGAMP than the PC7A polymer (molecular mass (MM) = 21 kDa). Second, cGAMP-induced conformational change of STING and subsequent oligomerization[20,21] may also occur faster than PC7A-induced STING condensate formation for immune activation. Finally, buffering of endosomal pH and disruption of endosomal membranes by PC7A deter STING degradation through the endosome–lysosome pathway. With the ability of PC7A to activate STING and cytosolic delivery of cGAMP, we demonstrate that cGAMP–PC7A NPs achieved rapid and sustained STING activation across 6–48 h in both MC38 tumours and draining lymph nodes (Supplementary Fig. 9), which enable an optimal time window for DC maturation and T-cell priming (normally requires 1–2 d)[52,53]. This is supported by the synergistic therapeutic outcomes of cGAMP–PC7A NPs in MC38 and TC-1 tumour treatment over single therapy alone.

A growing number of studies report the importance of STING pathway in cancer immunotherapy[12–17]. However, it is unclear whether STING activity in the cancer cells, immune cells or stromal cells have a more critical role in antitumour immunity. Our studies revealed the importance of host STING activity in cGAMP–PC7A NP therapy (Supplementary Fig. 12). Data also show that tumour growth inhibition is abolished by antibody blockade of CD8 T cells but not NK cells, indicating that CD8 T cells are the ultimate effector cells against tumours. We also show partial reduction of antitumour efficacy in DC-depleted mice, suggesting that additional immune cells (such as macrophages and B cells) or stromal cells (such as fibroblasts) may also contribute to the T-cell-mediated antitumour immunity. Further investigations are warranted to elucidate the contributions from other immune cell types or subset of immune cells (for example, tumour-resident CD103+ DCs)[11,54], which may help to identify key biomarkers for clinical translation.

In summary, this study highlights the use of a synthetic polymer to induce STING condensation for activation of an important innate immune pathway with spatiotemporal dynamics distinct from a natural STING ligand. Combination of polyvalent STING activation by PC7A with cell-intrinsic cGAMP stimulation further offers a synergistic and robust strategy to mount antitumour immunity for cancer immunotherapy.

## Methods

**Synthesis of polymers.** Monomers including 2-hexamethyleneiminoethyl methacrylate (C7A-MA), 2-(4-methylpiperidineleneimino)ethyl methacrylate (C6S1A-MA), 2-heptamethyleneiminoethyl methacrylate (C8A-MA), 2-diisopropylaminoethyl methacrylate (DPA-MA) and 2-ethylpropylaminoethyl methacrylate (EPA-MA) were synthesized according to previous publications[39,55]. PEG-*b*-PR copolymers were synthesized using an atom transfer radical polymerization method. Poly(ethylene glycol)-*b*-poly(2-hexamethyleneiminoethyl methacrylate) with 70 repeating units, that is, PC7A(70), is used as an example to illustrate the procedure. First, C7A-MA (1.5 g, 7 mmol), MeO-PEG$_{114}$-Br (0.5 g, 0.1 mmol, Sigma-Aldrich) and N,N,N′,N′′,N′′-pentamethyldiethylenetriamine (PMDETA, 21 μl, 0.1 mmol, Sigma-Aldrich) were dissolved in a mixture of 2-propanol (2 ml) and dimethylformamide (2 ml) in a Schlenk flask. Oxygen was removed by three cycles of freeze–pump–thaw, then CuBr (14 mg, 0.1 mmol, Alfa Aesar) was added under nitrogen protection. Polymerization was performed in vacuo at 40 °C overnight. After polymerization, the reaction mixture was diluted in tetrahydrofuran (10 ml), and then passed through a neutral Al$_2$O$_3$ column to remove the catalyst. The organic solvent was removed by rotary evaporation. The residue was dialysed in distilled water and lyophilized to obtain a white powder. After syntheses, the product was characterized using $^1$H NMR and gel permeation chromatography. The four other polymers, including PC6S1A, PC8A, PDPA and PEPA, were all synthesized with 70 repeating units. PC7A polymers with different repeating units were synthesized by adjusting the initial ratio of C7A-MA monomer over the MeO-PEG$_{114}$-Br initiator.

The synthesis of dye-conjugated copolymers was performed according to a similar procedure[39,55]. Primary amino groups (aminoethyl methacrylate or AMA-MA, Polysciences) were introduced into each polymer chain by controlling the feeding ratio of AMA-MA monomer to the initiator (3:1). After synthesis, PEG-*b*-(PR-*r*-AMA) was dissolved in dimethylformamide, and dye-*N*-hydroxylsuccinimidal ester was added (3 molar equivalences to the primary amino group, Lumiprobe). After overnight reaction, the copolymer was purified using ultracentrifugation (MM = 10 kDa cut-off) three times to remove free dye molecules. The product was lyophilized and stored at −80 °C.

**Preparation of micelle NPs.** Micelle NPs for cellular studies were prepared according to a solvent evaporation method as previously reported[39,55]. In brief, polymer (4 mg) was first dissolved in methanol (0.4 ml) and then added dropwise into distilled water (3.6 ml) under sonication. Methanol was removed by ultrafiltration (MW = 100 kDa cut-off) three times with fresh distilled water. Sterile PBS was added to adjust the concentration to 200 μM as a stock solution.

cGAMP-loaded NPs were prepared by mixing 2′3′-cGAMP in PC7A polymer solution containing 5% D-glucose at pH 4, and then adjusted to pH 7.4 using NaOH. After micelle formation, the NPs were analysed by dynamic light scattering to measure size and zeta potential, and transmission electron microscopy to analyse particle morphology. The cGAMP loading efficiency (>90%) was quantified using high-performance liquid chromatography.

 NATURE BIOMEDICAL ENGINEERING

**Expression, purification and labelling of recombinant STING proteins.** A human STING CTD (amino acid sequence 139–379) plasmid containing a $His_6$ tag encoded in the pET-SUMO vector (provided by Z. J. Chen, UT Southwestern) was used as a template to generate the E296A/D297A, D319A/D320A and E336A/E337A/E339A/E340A mutants using a Q5 site-directed mutagenesis kit (NEB). Overexpression of the WT or mutant protein was induced in *Escherichia coli* strain BL21/pLys with 0.8 mM isopropyl-β-D-thiogalactoside at 16 °C for 18 h. Bacterial cells were collected, suspended (50 mM Tris-Cl, 300 mM NaCl, 20 mM imidazole, pH 8.0) and disrupted by sonication on ice. Cellular debris was removed by centrifugation at 20,000*g* at 4 °C for 1 h. The supernatant was loaded onto a $Ni^{2+}$-nitrilotriacetate affinity resin (Qiagen). After 4 h incubation at 4 °C, the resin was rinsed three times with washing buffer (50 mM Tris-Cl, 1 M NaCl, 20 mM imidazole, pH 8.0). The SUMO tag was then removed by digesting the proteins using ULP1 SUMO protease at 4 °C overnight. Proteins were eluted with elution buffer (20 mM Tris-Cl, 50 mM NaCl, 20 mM imidazole, pH 7.5). Subsequently, the eluted proteins were analysed by size-exclusion chromatography using a Superdex 200 column (GE Healthcare), and the fractions were collected, concentrated and dialysed against a buffer containing 25 mM HEPES and 150 mM NaCl (pH 7.5)[43].

For dye conjugation, the protein solution was mixed with Cy5-NHS in $NaHCO_3$ (pH 8.4) at 4 °C overnight. Free dye molecules were removed using a desalting column (7 K, Thermo Fisher Scientific). Dye-labelled proteins were collected, concentrated and used in phase-separation studies.

**ITC analysis.** A MicroCal VP-ITC was used to measure the binding affinity between protein and polymer. STING dimer concentration was held at 12.5 μM and PC7A(70) at 10 μM. The titrations were performed at 20 °C in a buffer containing 25 mM HEPES and 150 mM NaCl (pH 6.5). Twenty-nine injections were performed in 3 min spacing time. The titration traces were integrated using NITPIC v.1.2.7, the curves were fitted using SEDFHAT v.15.2b and the figures were prepared using GUSSI v.1.4.2.

**Nile Red assay.** The Nile Red assay is used to study protein–protein interactions and interruptions in protein structure[37]. In brief, Nile Red (final concentration 5 μM, Thermo Fisher Scientific), STING dimer (2.1 μM) and PC7A (0 μM, 0.6 μM, 1.2 μM, 3 μM, 6 μM or 12 μM) were mixed for 4 h. Their maximum excitation wavelengths and fluorescence intensities were recorded on a fluorescence spectrophotometer (Hitachi F-7000 model).

**Phase condensation assay.** WT or mutant human STING CTD (Cy5-labelled) was mixed with PC7A polymers of varying repeating units in a 96-well glass plate (coated with mPEG-silane) at 25 °C. After 4 h, the mixture was centrifuged at 13,000*g* for 5 min, and the supernatant was transferred to another plate. Fluorescence intensity of the supernatant was measured using a plate reader (CLARIOstar). Data are representative of at least three independent measurements. The degree of condensation (*D*) was calculated using the following equation:

$$D_i = \frac{F_0 - F_i}{F_0}$$

where $F_i$ is the fluorescence intensity of the supernatant for a specific group *i*, and $F_0$ is the Cy5–STING intensity at the same concentration without PC7A addition.

For phase reversibility assays, STING CTD (Cy5-labelled) and PC7A polymer were first mixed. After condensate formation, the mixture was diluted ten times in pH 6.5 HEPES buffer, and shaken on a plate shaker for 24 h. The fluorescence intensity of the supernatant was measured, and reversibility (*R*) was calculated using following equation:

$$R_i = \frac{D_i - D_{R_i}}{D_i}$$

where $D_{R_i}$ was the new *D* value after 24 h recovery.

For microscopy examination, STING protein (Cy5-labelled) was mixed with PC7A polymer in a four-well glass chamber (Thermo Fisher Scientific; coated with mPEG-silane) at 25 °C, and images were acquired over a 140 s time course at intervals of 4 s using the built-in software (ZEN v.2.6) of the Zeiss 700 confocal laser scanning microscope. Size was calculated as the average of the longest and shortest axis of each condensate. The size distribution was plotted using GraphPad Prism 7.

**Animals and cells.** All of the animals were maintained at the animal facilities under specific-pathogen-free conditions and all animal procedures were performed with ethical compliance and approval by the Institutional Animal Care and Use Committee at the University of Texas Southwestern Medical Center. Female C57BL/6 mice (aged 6–8 weeks) were obtained from the UT Southwestern breeding core. Host *Tmem173*[−/−] C57BL/6 mice[56] were provided by Y.-X. Fu and CD11c-DTR transgenic C57BL/6 mice were purchased from the Jackson Laboratory. Mice were housed in a barrier facility under a 12 h–12 h light–dark cycle and maintained on standard chow (2916, Teklad Global). The temperature range for the housing room is 68–79 °F (average is around 72 °F) and the humidity range is 30–50% (average is around 50%).

GFP–STING MEFs (provided by N. Yan, UT Southwestern), and HEK293T (ATCC), B16F10 (ATCC), MC38 (ATCC), *Tmem173-KO* MC38 (provided by Y-X. Fu)[56], TC-1 (provided by T. C. Wu, John Hopkins University) cells were cultured in complete DMEM medium supplemented with 10% fetal bovine serum (FBS). THP1 cells (ATCC) were cultured in RPMI medium supplemented with 10% FBS and 0.05 mM β-mercaptoethanol (β-ME). All cells were grown at 37 °C in 5% $CO_2$. THP1 monocytes were differentiated into macrophages by phorbol 12-myristate 13-acetate (PMA, 150 nM, InvivoGen) before use.

In the cell mutagenesis assay, GFP-tagged full-length WT STING plasmid (provided by N. Yan) was used as a template to generate E296A/D297A, D319A/D320A and E336A/E337A/E339A/E340A mutants. HEK293T cells were transfected with lipofectamine 2000 (Invitrogen) carrying full-length WT or mutant STING-GFP plasmid for 24 h and allowed to recover for 12 h before use. WT or R232H THP1 reporter cells were purchased from Invitrogen. R238A/Y240A and single or dual Q273A/A277Q HeLa mutants (provided by Z. J. Chen)[20,21] were used as cGAMP-resistant STING mutant cells.

**Microscopy.** Cells were grown in a four-well glass chamber and treated with cGAMP or PC7A polymer for the indicated time. In the STING degradation assay, LysoTracker Red DND-99 (Thermo Fisher Scientific) was used to stain lysosomes in live cells. In the STING trafficking assay, cells were fixed in 4% paraformaldehyde, then permeabilized and stained for ER (Calnexin, 1:200, Abcam), ERGIC (p58, 1:1,000, Sigma-Aldrich), Golgi (GM130, 1:50, BD Biosciences) or p-TBK1 (Ser 172, 1:50, Cell Signaling Technologies) using an immunofluorescence kit (Cell Signaling Technologies). Samples were mounted in prolong gold antifade with DAPI stain (Thermo Fisher Scientific) and imaged using the built-in software (ZEN 2.6) of the Zeiss 700 confocal laser scanning microscope with a ×63 oil-immersion objective. ImageJ v.1.52d was used to quantify colocalization using the Pearson's correlation coefficient. Data are representative of at least 20 cells. In the inhibitor assay, cells were pretreated with BFA (10 μM, Selleckchem) for 1 h before cGAMP/PC7A addition.

**FRAP experiments.** The FRAP method is a versatile tool for determining the diffusion and exchange properties of biomacromolecules[57]. Both in vitro and cellular FRAP experiments were performed using a Zeiss 700 confocal laser scanning microscope at 25 °C. In a typical procedure, a 2-μm-diameter spot in the condensation was photobleached with 100% laser power for 5 s using a 633 nm laser. Images were acquired over a 150 s time course at intervals of 4 s. Fluorescence intensity of the region of interest was corrected by an unbleached control region and then normalized to the prebleached intensity of the region of interest. At least five biologically independent samples were measured. The mean intensity of the bleached spot was fit to a single exponential model[32] using Graph Pad Prism 7 software.

**Western blot analysis.** All solutions were purchased from Bio-Rad and antibodies against STING (1:1,000), p-STING (S366, 1:1,000), p-TBK1 (Ser 172, 1:1,000) and p-IRF3 (Ser 369, 1:1,000) were obtained from Cell Signaling Technologies. In brief, cells were lysed in SDS sample buffer (with protease and phosphatase inhibitor cocktail) and heated for denaturation. Supernatant was loaded onto a 4–15% Mini-PROTEAN gel (Bio-Rad), and run at 50 V for 20 min followed by 100 V for 60 min. Electrotransfer was performed using 100 V for 60 min on ice. After transfer, the membrane was blocked either in 5% non-fat milk or BSA (phosphorylated protein) for 1 h at room temperature, and incubated with primary antibodies overnight at 4 °C. Goat anti-mouse or goat anti-rabbit IgG HRP-linked secondary antibody (1:3,000, Bio-Rad) was used for 1 h at room temperature before detection on X-ray film (GE Healthcare). The membrane was stripped in stripping buffer for 30 min and reused for β-actin (Sigma-Aldrich) detection.

**RT–qPCR.** Total RNAs were extracted from cells or human tissues using the RNeasy mini kit (Qiagen). RNA quantity and quality were confirmed using the NanoDrop (DeNovix DS-11) system. Genomic DNA was removed and cDNA was synthesized using an iScript gDNA clear cDNA synthesis kit (Bio-Rad). Bio-Rad SsoAdvanced universal SYBR green supermix and CFX connect real-time system were used for PCR analysis. Results were corrected by *ACTB* or *GAPDH* in Excel Office 365 and plotted in Graph Pad Prism 7. The DNA primers used were as follows: mouse *Ifnb1*: ATGAGTGGTGGTTGCAGGC, TGACCTTTCAAATGCAGTAGATTCA; mouse *Cxcl10*: GGAGTGAA GCCACGCACAC, ATGGAGAGAGGCTCTCTGCTGT; mouse *Actb*: ACACCCGCCACCAGTTCGC, ATGGGGTACTTCAGGGTCAGGATA; human *IFNB1*: GTCTCCTCCAAATTGCTCTC, ACAGGAGCTTCTGACACTGA; human *CXCL10*: TGGCATTCAAGGAGTACCTC, TTGTAGCAATGATCTCA ACACG; human *ACTB*: GGACTTCGAGCAAGAGATGG, AGGAAGG AAGGCTGGAAGAG; human *GAPDH*: ATGACATCAAGAAGGTGGTG, CATACCAGGAAATGAGCCTTG.

**Evaluation of STING activation in tumour-bearing mice.** Mice were subcutaneously inoculated with MC38 cells ($1 \times 10^6$) into the right flank. One intratumoural injection of different agents (50 μl of 5% glucose, 50 μg PC7A polymer, 2.5 or 50 μg cGAMP or a formulation with 2.5 μg cGAMP in 50 μg PC7A

NPs) was performed when the tumour size reached $100 \pm 20$ mm³. Mice were euthanized at different time points after injection, and tumours and draining lymph nodes were collected. Total RNAs were extracted by TRIzol (Invitrogen), and the expression of interferon-stimulated genes (*Ifnb1* and *Cxcl10*) was measured using quantitative PCR with reverse transcription (RT–qPCR).

**Safety studies.** Mice were subcutaneously inoculated with MC38 cells ($1 \times 10^6$) into the right flank. Intratumoural injections of different agents (50 µl of 5% glucose, 50 µg PC7A polymer, 2.5 or 50 µg cGAMP or a formulation with 2.5 µg cGAMP in 50 µg PC7A NPs) were performed when the tumour size reached ~50 mm³ (around day 6). Two additional injections were performed on days 9 and 12. One day after the final administration, 1 ml of blood sample was collected from each mouse without heparinization and then centrifuged at 4,000 r.p.m. for 5 min to obtain serum. The activities of alanine aminotransferase (ALT), aspartate aminotransferase (AST) and urea were measured using specific kits (Abcam, 105134, 105135, 83362, respectively). The systemic concentration of interleukin-10 was measured using an enzyme-linked immunosorbent assay (Invitrogen, 88-7105-22). Statistical analysis was performed using GraphPad Prism 7.

**Tumour therapy experiments.** Mice were subcutaneously inoculated with MC38 cells ($1 \times 10^6$) or TC-1 cells ($1 \times 10^5$) into the right flank. Tumour size was measured every 2 d or 3 d using digital callipers, and tumour volume was calculated as $0.5 \times \text{length} \times \text{width}^2$. On reaching sizes of ~50 mm³, tumours were injected with different STING agonists (50 µl of 5% glucose, 50 µg PC7A polymer, 2.5 µg or 50 µg cGAMP, or 2.5 µg cGAMP in 50 µg PC7A NPs), and some of the groups were intraperitoneally injected with 200 µg depletion antibodies (anti-mCD8α, BioXcell, BP0117 or anti-mNK1.1, BioXcell, BP0036) or 200 µg checkpoint inhibitors (anti-mPD-1, BioXcell, BE0146) every 3 d for comparison or synergy evaluation. For systemic DC depletion, CD11c-DTR transgenic mice were injected intraperitoneally with 100 ng diphtheria toxin (Sigma-Aldrich) every 3 d after tumour inoculation. Mice were injected three times in the MC38 model and four times in the TC-1 model with STING agonist treatments spaced 3 d apart. Mice were euthanized at a tumour burden endpoint of 2,000 mm³. Statistical analysis was performed using GraphPad Prism 7.

**Evaluation of STING activation in resected human tissues.** Patients provided consent for the use of biospecimens for research as approved by the UT Southwestern Institutional Review Board. Freshly resected human tissues (squamous cell carcinoma from the base/lateral of tongue, cervical tumour tissues and a sentinel lymph node) were rinsed and divided into several sections (1–5 mm³) using a scalpel, followed by injection at multiple sites using 5% glucose control, free cGAMP (80 ng), PC7A polymer (50 µg) or cGAMP–PC7A NPs (80 ng cGAMP in 50 µg PC7A NPs) in 5% glucose solution within 30 min of resection. Each section was cultured in 0.5 ml RPMI 1640 medium (supplemented with 10% heat-inactivated human serum, 1% insulin-transferrin-selenium, 1% GlutaMAX, and 1% penicillin–streptomycin) in a 24-well plate for 24 h. RNA was isolated and RT–qPCR was performed as previously described. For CD45 selection, tumour tissues were first digested by 1 mg ml⁻¹ collagenase IV and 0.2 mg ml⁻¹ DNase I (Sigma-Aldrich) for 45 min at 37 °C, then passed through a 70 µm nylon cell strainer to obtain single cells. CD45⁺ leukocytes and CD45⁻ cell populations were collected using magnetic separation using CD45 TIL microbeads and MS columns (Miltenyi Biotec) according to the manufacturer's instructions before RT–qPCR analysis.

**Reporting Summary.** Further information on research design is available in the Nature Research Reporting Summary linked to this article.

## Data availability
The main data supporting the results in this study are available within the paper and its Supplementary Information. All data generated in this study, including source data and the data used to generate the figures, are available at figshare (https://doi.org/10.6084/m9.figshare.13356464).

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

## Acknowledgements

We thank Z. J. Chen, M. Du, C. Zhang, C. Han and N. Yan for sharing plasmids, STING mutant cell lines and STING knockout animals; M. Rosen for discussion on biomolecular condensation; S. Y. Wu and H. T. Lai for the construction of STING mutant plasmids; Y. Li and L. You for polymer synthesis and characterization; C. Chen for computational modelling; S. Tso for ITC experiments; J. Lea for procurement of cervical tumour tissues; and G. Huang, T. Huang, Z. Sun and Z. Liu for discussions. This work was supported by a grant from the National Institutes of Health (U54 CA244719) and Mendelson-Young Endowment in Cancer Therapeutics to J.G.

## Author contributions

S.L. designed and performed the majority of the experiments, analysed the data and wrote the first draft. M.L. made the initial observation of sustained STING activity in live cells by PC7A. Z.W. generated dye-labelled polymers/proteins and assisted with confocal imaging analysis. Q.F. and J.Wilhelm assisted with polymer synthesis and characterization. X.W. developed the formulation of cGAMP-loaded PC7A NPs. W.L. and J.Wang assisted with animal studies and cytokine analysis. Y.-x.F. assisted with STING-knockout and immune-cell-dependent studies. A.C. and B.D.S. performed experiments in human tissues. H.Y. assisted with the experimental design on STING–PC7A interactions and biomolecular condensation. J.G. supervised all of the experiments and revised the final manuscript.

## Competing interests

B.D.S. and J.G. are scientific co-founders and advisors of OncoNano Medicine, Inc.

## Additional information

**Correspondence and requests for materials** should be addressed to J.G.

**Peer review information** Peer reviewer reports are available.

# naturereseach

# Reporting Summary

Nature Research wishes to improve the reproducibility of the work that we publish. This form provides structure for consistency and transparency in reporting. For further information on Nature Research policies, see Authors & Referees and the Editorial Policy Checklist.

## Statistics

For all statistical analyses, confirm that the following items are present in the figure legend, table legend, main text, or Methods section.

| n/a | Confirmed | |
|---|---|---|
| ☐ | ☒ | The exact sample size (*n*) for each experimental group/condition, given as a discrete number and unit of measurement |
| ☐ | ☒ | A statement on whether measurements were taken from distinct samples or whether the same sample was measured repeatedly |
| ☐ | ☒ | The statistical test(s) used AND whether they are one- or two-sided<br>*Only common tests should be described solely by name; describe more complex techniques in the Methods section.* |
| ☒ | ☐ | A description of all covariates tested |
| ☒ | ☐ | A description of any assumptions or corrections, such as tests of normality and adjustment for multiple comparisons |
| ☐ | ☒ | A full description of the statistical parameters including central tendency (e.g. means) or other basic estimates (e.g. regression coefficient) AND variation (e.g. standard deviation) or associated estimates of uncertainty (e.g. confidence intervals) |
| ☐ | ☒ | For null hypothesis testing, the test statistic (e.g. *F*, *t*, *r*) with confidence intervals, effect sizes, degrees of freedom and *P* value noted<br>*Give P values as exact values whenever suitable.* |
| ☒ | ☐ | For Bayesian analysis, information on the choice of priors and Markov chain Monte Carlo settings |
| ☒ | ☐ | For hierarchical and complex designs, identification of the appropriate level for tests and full reporting of outcomes |
| ☐ | ☒ | Estimates of effect sizes (e.g. Cohen's *d*, Pearson's *r*), indicating how they were calculated |

*Our web collection on statistics for biologists contains articles on many of the points above.*

## Software and code

Policy information about availability of computer code

| | |
|---|---|
| Data collection | Confocal images were taken with the built-in software (ZEN 2.6) of the Zeiss 700 confocal laser scanning microscope. mRNA expressions were measured by the Bio-Rad CFX Connect Real-Time PCR Detection System. Luciferase activities and fluorescent intensity of phase separation assays were measured with the CLARIOstar plate reader. Titration curves were generated with the MicroCal VP-ITC system. |
| Data analysis | Statistical analyses were performed with Graph Pad Prism 7 and Excel Office 365. For co-localization analyses, the Pearson's correlation coefficient was calculated via ImageJ 1.52d. For the ITC assay, titration traces were integrated by NITPIC 1.2.7, the curves were fitted by SEDPHAT 15.2b, and the figures were prepared by using GUSSI 1.4.2 software. |

For manuscripts utilizing custom algorithms or software that are central to the research but not yet described in published literature, software must be made available to editors/reviewers. We strongly encourage code deposition in a community repository (e.g. GitHub). See the Nature Research guidelines for submitting code & software for further information.

## Data

Policy information about availability of data

All manuscripts must include a data availability statement. This statement should provide the following information, where applicable:
- Accession codes, unique identifiers, or web links for publicly available datasets
- A list of figures that have associated raw data
- A description of any restrictions on data availability

The main data supporting the results in this study are available within the paper and its Supplementary Information. All data generated in this study, including source data and the data used to generate the figures, are available from figshare with the identifier https://doi.org/10.6084/m9.figshare.13356464.

# Field-specific reporting

Please select the one below that is the best fit for your research. If you are not sure, read the appropriate sections before making your selection.

☒ Life sciences ☐ Behavioural & social sciences ☐ Ecological, evolutionary & environmental sciences

For a reference copy of the document with all sections, see nature.com/documents/nr-reporting-summary-flat.pdf

# Life sciences study design

All studies must disclose on these points even when the disclosure is negative.

| | |
|---|---|
| Sample size | For RT-PCR, luciferase activity, and quantification of the degree of phase separation, each sample was performed three times independently. For co-localization analyses, 20 confocal images were randomly taken and the Pearson's correlation coefficient was calculated from all the cells in these images. For quantification of STING puncta in lysosomes, 100 puncta were randomly taken from 20 cells and this process was repeated three times. For the FRAP assay, five condensates were randomly picked, and bleaching was performed at the central site of each condensate. For the quantification of condensate size, 50 condensates were randomly taken and their size was calculated as the average of the longest axis and the shortest axis. For the in vivo studies, animals in each model were randomly divided with 5–7 mice per group. For the human-tissue assay, each SLN or SCC-BOT tissue was divided into 16 sections before treatment (4 in each group); the cervical tumor tissue was divided into 8 sections because of their small size (2 in each group). The sample sizes were selected on the basis of a power analysis of results from preliminary experiments, and were consistent with those used in similar reports in the literature, which indicates that the sample sizes are not only sufficient to obtain desirable significance level (< 0.01) and power (> 90%), but also able to generate highly reproducible results with biological replicates. |
| Data exclusions | No data were excluded. |
| Replication | RT-PCR and luciferase-activity measurements were performed three times, biologically independently. Imaging experiments (that is, confocal and western blot) were performed at least three times biologically independently, with all replicates generating similar results. In the in vivo studies, 5–7 mice per group were used and no replicates were performed. In human-tissue assays, two technical replicates were performed for each section. |
| Randomization | For the quantification of condensates or cells with phenotype of interest, images were randomly taken throughout the chamber of each sample. For in vitro cell-based RT-PCR, luciferase-activity measurement, and western-blot assay, all cells under well-controlled conditions were analysed equally; therefore, no randomization was necessary. For the animal study and the human-tissue assay, animals and divided sections were randomly allocated into each group. |
| Blinding | In all phase-separation assays and cell-based experiments, data collection and analysis were performed in a blinded manner because each sample was only identified with a number that didn't show any information about the treatment administrated. In the animal studies, true blinding was not possible for the administration of treatment owing to practical reasons, but the appropriate controls were present. The human-tissue assay was performed by independent researchers who were blinded as to treatment-group assignment. |

# Reporting for specific materials, systems and methods

We require information from authors about some types of materials, experimental systems and methods used in many studies. Here, indicate whether each material, system or method listed is relevant to your study. If you are not sure if a list item applies to your research, read the appropriate section before selecting a response.

## Materials & experimental systems

| n/a | Involved in the study |
|---|---|
| ☐ | ☒ Antibodies |
| ☐ | ☒ Eukaryotic cell lines |
| ☒ | ☐ Palaeontology |
| ☐ | ☒ Animals and other organisms |
| ☐ | ☒ Human research participants |
| ☒ | ☐ Clinical data |

## Methods

| n/a | Involved in the study |
|---|---|
| ☒ | ☐ ChIP-seq |
| ☒ | ☐ Flow cytometry |
| ☒ | ☐ MRI-based neuroimaging |

## Antibodies

| | |
|---|---|
| Antibodies used | Rabbit Monoclonal anti-STING (13647), Rabbit Monoclonal anti-Phospho-STING (S366) (19781), Rabbit Monoclonal anti-Phospho-TBK1/NAK (Ser172) (5483), and Rabbit Monoclonal anti-Phospho-IRF-3 (Ser396) (4947) were purchased from Cell Signaling. Mouse Monoclonal anti-β-actin (A5441) and Rabbit Polyclonal anti-ERGIC-53/p58 - Cy3 (E6782) were purchased from Sigma Aldrich. Mouse Monoclonal anti-Calnexin - AF647 (ab202572) was from Abcam. Mouse Monoclonal anti-GM130 - AF647 (558712) was from BD Biosciences. Goat Anti-Mouse IgG (H+L) - HRP (1721011) and Goat Anti-Rabbit IgG (H+L) - HRP (1706515) were from Bio-Rad. Anti-mouse PD-1 (BE0146), CD8α (BP0117), and NK1.1 (BP0036) were from Bio X Cell. |

| Validation | All antibodies are commercially available. Each antibody was validated for species and application, as appropriate, according to the manufacturer's website, and as supported by relevant citations on their product pages. |

# Eukaryotic cell lines

Policy information about cell lines

| Cell line source(s) | The cell lines, including HEK293T, Hela, THP1, B16 and MC38 cells, were obtained from ATCC. WT and R232H STING reporter cells were abtained from InvivoGen. ISG-THP1 cells, R238A/Y240A and single or dual Q273A/A277Q Hela mutants cells were constructed and provided by Z.J.C. (Nature 567.7748 (2019): 394–398). STING-GFP MEF cells were provided by N.Y. (Cell host & microbe 18.2 (2015): 157–168). Tmem173-KO MC38 cells were provided by Y-X. F. (Nature immunology 21.5 (2020): 546–554). TC-1 cells were provided by T. C. Wu (John Hopkins University). ISG-THP1 cells are commercially available from InvivoGen and TC-1 cells are commercially available from ATCC. |
|---|---|
| Authentication | HEK293T, Hela, THP1, B16, and MC38 cells were authenticated by ATCC. WT and R232H STING reporter cells were authenticated by InvivoGen. No further authentication was performed. |
| Mycoplasma contamination | The cell lines used in this study were free of mycoplasma contamination, according to results from the e-Myco Mycoplasma PCR Detection Kit (Bulldog Bio), and were regularly maintained with Normocin. |
| Commonly misidentified lines (See ICLAC register) | No commonly misidentified cell lines were used. |

# Animals and other organisms

Policy information about studies involving animals; ARRIVE guidelines recommended for reporting animal research

| Laboratory animals | Six-to-eight week-old female C57Bl/6 mice were purchased from the UT Southwestern breeding core. Tmem173-/- C57BL/6 mice were provided by Y-X. F. CD11c-DTR transgenic C57BL/6 mice were purchased from the Jackson Laboratory. Mice were housed in a barrier facility with a 12-h light/dark cycle and maintained on standard chow (2916 Teklad Global). The temperature range for the housing room is 68–79 ºF (average is around 72 ºF) and the humidity range is 30–50% (average is around 50%). |
|---|---|
| Wild animals | The study did not involve wild animals. |
| Field-collected samples | The study did not involve samples collected from the field. |
| Ethics oversight | The handling of mice and the experiments with them were conducted under federal, state, and local guidelines and with approval from the Institutional Animal Care and Use Committee at the University of Texas Southwestern Medical Center. |

Note that full information on the approval of the study protocol must also be provided in the manuscript.

# Human research participants

Policy information about studies involving human research participants

| Population characteristics | Surgically resected human tissues were obtained from four individual patients. #1. 55-year-old; male; diagnosis: squamous cell carcinoma of the base of tongue, stage I; treatment: adjuvant radiation and no chemotherapy; current status: clinically and radiographically without evidence of disease. #2. 29-year-old; female; diagnosis: squamous cell carcinoma of the right lateral tongue, stage I; treatment: adjuvant radiation and no chemotherapy; current status: no evidence of persistent disease. #3. 56-year-old; female; diagnosis: cervical cancer, stage IVB; treatment: chemotherapy including cisplatin and paclitaxel; current status: remission for one year. #4. 72-year-old; male; diagnosis: squamous cell carcinoma of the left piriform sinus, stage IVA; treatment: adjuvant radiation and cisplatin; current status: no recurrent neck mass or pathologically enlarged lymphadenopathy. |
|---|---|
| Recruitment | Patients were invited to participate by one of the investigators or a designee of one of the investigators. If they were willing to participate, the investigator or their designee reviewed the consent form and obtained informed consent. All patients with planned surgery were consented to the tissue-collection protocol. Patients were approached in the clinics by study personnel without relation to age, race, gender, disease stage or prior therapies. Tissue collection and formulation injection were performed by independent researchers who were blinded as to treatment-group assignment. There was no self-selection or any other bias. |
| Ethics oversight | UT Southwestern Institutional Review Board (IRB number 092013-032). |

Note that full information on the approval of the study protocol must also be provided in the manuscript.

