## [Peer Review File · Nature Biomedical Engineering]

Prolonged activation of innate immune pathways by a polyvalent STING agonist

Corresponding author: Jinming Gao

Editorial note

This document includes relevant written communications between the manuscript's corresponding author and the editor and reviewers of the manuscript during peer review. It includes decision letters relaying any editorial points and peer-review reports, and the authors' replies to these (under 'Rebuttal' headings). The editorial decisions are signed by the manuscript's handling editor, yet the editorial team and ultimately the journal's Chief Editor share responsibility for all decisions.

Any relevant documents attached to the decision letters are referred to as **Appendix #**, and can be found appended to this document. Any information deemed confidential has been redacted or removed. Earlier versions of the manuscript are not published, yet the originally submitted version may be available as a preprint. Because of editorial edits and changes during peer review, the published title of the paper and the title mentioned in below correspondence may differ.

A description of the work was initially submitted as a presubmission enquiry. Reviewer #3 was unavailable to re-review the manuscript.

Correspondence

Wed 12/02/2020

Decision on Presubmission Enquiry nBME-20-0300-PE

Dear Dr Gao,

Thank you for submitting to *Nature Biomedical Engineering* your Presubmission Enquiry, "A polyvalent STING agonist prolongs innate activation against cancer".

I appreciate that the longer activation of the STING pathway with respect to cGAMP by your polyvalent polymer has legs, and that using PC7A in combination therapies with immune checkpoint inhibitors could be a winning strategy. Now, because in your 2017 paper published in *Nature Nanotechnology* there was some preliminary evidence of direct binding between STING and PC7A, I anticipate that, in addition to the mechanistic evidence, our editorial assessment will largely focus on the performance outcomes. In such cases we typically look for at least two animal tumour models, with one of them orthotopic, and long-term follow-ups and survival curves. Also, evidence of favorable outcomes with i.v. delivery would be preferable.

We would thus be happy to assess the manuscript when you have it ready. When so, please upload the manuscript files as well as our reporting summary and policy checklist. (Please note that these forms are dynamic PDF files that can only be properly visualized and filled in by using Acrobat Reader.) Both documents are aimed at ensuring good reporting standards and at easing the interpretation of results, and will be available to the reviewers. Should the manuscript be eventually published, the reporting summary will be attached to the published PDF of the paper and will also be available as supplementary information. More information is available on the editorial policies page.

Best wishes,

Pep

Pep Pàmies
Chief Editor, Nature Biomedical Engineering

Wed 12/02/2020

Re: Decision on Presubmission Enquiry nBME-20-0300-PE

Dear Pep,

Thank you for the positive news and thoughtful response. We have antitumor efficacy data in two animal tumor models, as well as STING activation results in patient tumors and lymph nodes to mitigate potential risks for clinical translation. We will follow your suggestion to prepare the reporting summary and policy checklist, and submit the manuscript in due course.

Warmest regards,

Jinming

Wed 19/02/2020

Re: Decision on Presubmission Enquiry nBME-20-0300-PE

Dear Dr Gao,

Thanks for submitting the full manuscript.

My colleague João Duarte and I have discussed the work. We feel that despite that Fig. 5 shows only delayed tumour growth with intratumoural delivery, the substantial mechanistic data in Figs. 3 and 4 and the activation in human tissues (Fig. 6) overall makes for a compelling manuscript (that is, editorially we have given less weight to the performance outcomes). We will send the manuscript for external peer review.

João will handle the manuscript.

Thanks,

Pep

Thu 12/03/2020

Decision on Article NBME-20-0300A

Dear Dr Gao,

Thank you again for submitting to *Nature Biomedical Engineering* your Article, "A polyvalent STING agonist prolongs innate activation against cancer". The manuscript has been seen by three experts, whose reports you will find at the end of this message. You will see that although the reviewers have some good words for the work, they articulate concerns about the degree of advance that this study represents over relevant published work, and in this regard provide useful suggestions for improvement. We hope that with significant further work you can address the criticisms, increase the level of significance of the study, and convince the reviewers of its merits. In particular, we would expect that a revised version of the manuscript provides:

- * a thorough assessment of the potential for dose-limiting toxicity of the formulation tested, as highlighted by all reviewers;
- * a clarification of the discrepancies in the doses used between PC7A and cGAMP, including testing of higher cGAMP doses as requested by reviewers #2 and #3;
- * a complete description of the formulation used in the manuscript (which differs from the PC7A formulation previously described);
- * a deeper insight into the mechanism of action of PC7A in terms of STING activation (reviewer #1 suggests experiments in STING-KO mice) and into the effects on multiple cells of the immune system where the STING pathway is active.

When you are ready to resubmit your manuscript, please upload the revised files, a point-by-point rebuttal to the comments from all reviewers, the (revised, if needed) reporting summary, and a cover letter that explains the main improvements included in the revision and responds to any points highlighted in this decision.

Please follow the following recommendations:

- * Clearly highlight any amendments to the text and figures to help the reviewers and editors find and understand the changes (yet keep in mind that excessive marking can hinder readability).
- * If you and your co-authors disagree with a criticism, provide the arguments to the reviewer (optionally, indicate the relevant points in the cover letter).
- * If a criticism or suggestion is not addressed, please indicate so in the rebuttal to the reviewer comments and explain the reason(s).
- * Consider including responses to any criticisms raised by more than one reviewer at the beginning of the rebuttal, in a section addressed to all reviewers.
- * The rebuttal should include the reviewer comments in point-by-point format (please note that we provide all reviewers will the reports as they appear at the end of this message).
- * Provide the rebuttal to the reviewer comments and the cover letter as separate files.

We hope that you will be able to resubmit the manuscript within 15 weeks from the receipt of this message. If this is the case, you will be protected against potential scooping. Otherwise, we will be happy to consider a revised manuscript as long as the significance of the work is not compromised by work published elsewhere or accepted for publication at *Nature Biomedical Engineering*.

We hope that you will find the referee reports helpful when revising the work. Please do not hesitate to contact me should you have any questions.

Best wishes,

João

Dr João Duarte

Reviewer #1 (Report for the authors (Required)):

This manuscript by Li et al., describes the unique role of a synthetic polymer, PC7A, to induce STING condensation through polyvalent interactions for activation of immune responses. The authors found that this PC7A polymer binds to a distinct STING surface site from the cGAMP-binding pocket, thus retaining STING activation in cGAMP-resistant clinical variants. The authors further claimed and demonstrated that combination of cGAMP with PC7A nanoparticles could offer a synergistic means to prolong animal survival (in vivo mouse model) and to activate STING in fresh human tissues (ex vivo). The work is based on a previous publication by the authors (Nature Nanotech, 2017, 12, 648) that initially reported the use of PC7A in STING activation. The current work attempted to develop an in-depth understanding of how PC7A uniquely activates the STING pathway and how it can be used with cGAMP to achieve a synergistic treatment effect. Overall, this is a solid study that elucidates the mechanistic role of a previously reported PC7A polymer in acting STING. However, the innovative aspect of the current work seems incremental and falls a bit short for the intended journal. In addition, there are a number of critical issues that the authors should consider.

1. The authors claimed STING-dependent activation of immunity by PC7A, but some important results are missing to support the unique contribution of PC7A-induced STING pathways. At least, the authors should consider using STING and IFN knockout mice to confirm the importance of STING pathway.
2. Activation of STING could also promote dendritic cells, nature killer cells and cytotoxic T cells for immune response, however, none of them was examined in the current manuscript. The authors should assess the contribution of these cells in the PC7A induced antitumor efficacy.
3. Drug dosage plays a pivotal role in their therapeutic efficacy. In the current studies, the authors chose to use a much lower dose of cGAMP (2.5 µg) than PC7A (50 µg) in animal studies (see Figure 5). In Figures 5a and 6, PC7A alone works better for cGAMP alone, while in Figure 5d and 5e, cGAMP works more effectively. Some explanations are necessary. What limits the dose of cGAMP? Do different ratios of combining the two offer different treatment outcomes?
4. In the same vein, given that the STING agonists often show dose-limiting toxicity (as claimed by the authors in the Introduction part), the safety and immune-related side effects should be assessed.
5. A minor point: survived animals should be monitored for at least several weeks after the death of last reported mice (figure 5e) to confirm no tumor recurrence.
6. Figure 1 shows both cGAMP and PC7A could promote STING puncta formation rapidly, however, the expression of IFNs in cGAMP is much quicker than PC7A. Why? The authors should consider discussing their time difference in immune response.
7. The PC7A polymer has been reported previously. However, when used for a new formulation, the nanocharacteristic could vary. The authors should provide characterization results of the PC7A/cGAMP particles (TEM, DLS, Zeta potential, etc). Also, given the high-water solubility of cGAMP, how could the authors achieve a high loading efficiency of >90% when preparing PC7A nanoparticles in aqueous solution?

Reviewer #2 (Report for the authors (Required)):

Summary

The manuscript describes the characterization of a novel STING-binding polymer that results in STING activation through a novel mechanism. The manuscript provides clear data showing direct interaction with STING resulting in phase shifts and activation of downstream signaling components and identifies a novel binding region. The study of STING as a therapeutic target has thus far focused on variations of the endogenous ligand, and this work represents a technological advance that has the potential to improve targeting of this pathway. There remain some issues in the approach when the drug is used as a therapy that limit the conclusions. For example, the central rationale for developing new ligands is that the current ligands exhibit dose limiting toxicities and haven't translated well to humans. This drug appears to work through identical pathways and data is not provided to suggest that this will avoid the same limitations.

Major issues.

1. The authors do not adequately explain why the combination of PC7A and the conventional ligand cGAMP are required for therapy. The manuscript suggests that 'cell intrinsic and noncanonical mechanisms' are involved, as well as increased delivery of cGAMP. However, these seem unnecessary in the in vitro studies

for PC7A action. Additional studies are needed to explain whether dual STING ligand treatment is superior in vivo through some mechanism of synergy or whether simply increasing the amount of STING ligands present results in improved outcomes.

2. In the described in vivo studies the high dose of cGAMP is 2.5µg, which is below its effective dose, so of course shows weaker effects than PC7A on cytokine release and in tumor control. Most preclinical studies use cyclic dinucleotides in the 25-100µg range, and that dropping below 10µg results in a loss of efficacy. No titration information is provided for PC7A, but uses a base dose of 50µg so it is plausible that this has very similar dose-response kinetics in vivo to the endogenous ligands. There is not enough information presented to suggest that this is an improved drug.

3. The title is only supported by the initial in vitro data on STING activation. There is no information provided to suggest that prolonged innate activation contributes to the responses against cancer, since only early peak cytokine responses are assessed. Additional experiments are necessary to determine whether PC7A exhibits a prolonged STING activation in vivo.

4. The models do not address the issue of dose limiting toxicity, which is one of the central rationales for the development of this drug.

Minor issues

1. Prior studies have shown that cyclic dinucleotides activate IFN secretion from mouse and patient tumor explants analyzed ex vivo (PMID: 30224374). The authors should explain why cyclin dinucleotides are not effective in their experiments.

2. Prior studies have shown synergy between STING ligands and anti-PD1 treatment in murine models (PMID: 27821498; PMID: 28483787). The authors should cite this work in their manuscript.

3. The fact that STING ligands dominantly activate non cancer cells in the tumor stroma rather than cancer cells (Figure 6) presents an undiscussed confounding issue. In particular, the mixed use of THP1 (monocytic) versus 293 and fibroblasts that are used in vitro compared to the murine cell lines that are used in vivo. These may have distinct expression of STING pathway genes and STING ligand pumps which make it difficult to extrapolate in vitro to in vivo effects. The role of cancer cell versus host STING activation would be better supported if the authors reported on STING expression by the mouse cancer cells used in the in vivo study, or used STING knockout mice or STING knockout cancer cells in vivo. However, a discussion of this issue would be sufficient.

Reviewer #3 (Report for the authors (Required)):

Li et al has shown that PC7A activate delayed STING activation compare to natural STING agonist cGAMP. STING activation though PC7A stays for more time compare to cGAMP activation. Though authors have reported a novel mechanism, but study is missing some of important controls. Authors claim that PC7A may have therapeutic implication in cGAMP nonresponsive patient. For best of my knowledge cGAMP is not being investigated in clinic. ADU-S100 is known STING and being investigated in clinic was not used in this study for comparison.

Major comment:

1- One of the benefit of small molecule STING agonist is its short half-life in vivo which is prepared for safety of the patients. Authors should discuss how PC7A can be used in patients. If a low dose of PC7A therapy can improve T cell response (reported by Chandra et al. 2014 and Sivick et al 2018), please discuss it.

2- 50ug of PC7A was tested to compare with 0.5ug and 2.5 ng of cGAMP or 50ug PC7A vs 80ng aGAMP. Please discuss why low does of cGAMP were used to compare. If optimum cGAMP was titrated for in vitro and in vivo assays. Please discuss rationale for using low doses of cGAMP.

Minor:

1-please mention, how many time each experiments were repeated and what is represented in figures.

2-Please also discuss , what authors think about delayed STING activation by PC7A compared to c-GAMP, as cytokine was mRNA activation seen after 12 hours of treatment

Mon 23/09/2020

Reviewer reports for Article NBME-20-0300B

Dear Dr Gao,

Thank you again for your manuscript "A polyvalent STING agonist prolongs innate activation against cancer". For your information, we have received two reports (included at the end of this message), but one reviewer has withdrawn from assessing the manuscript. In any case, in view of your response to the initial comments and to the two latest reports I am happy to tell you that we would be happy to, in principle, accept the manuscript for publication.

We will now need to carry out a number of checks to be able to provide you with the information that you will need to bring the manuscript into shape for production.

Best wishes,

João

Dr João Duarte
Senior Editor, Nature Biomedical Engineering

Reviewer #1 (Remarks to the Author):

The authors have adequately and satisfactorily addressed the previous raised comments.

Reviewer #2 (Remarks to the Author):

The manuscript is a revised response to review addressing all of the key points raised by this reviewer. This reviewer is particularly impressed with the response to questions of STING agonist dose, duration of effect, and the experiments in the STING knockout mice. These clearly show that the drug acts through STING in host cells in the tumor, provides a sustained STING activation, resulting in improved mechanisms of response. These data add to the already strong manuscript, and help us interpret the data and why this is new and improved.

No significant issues.

Mon 26/10/2020

Decision on Article NBME-20-0300B

Dear Dr Gao,

Thank you for your revised manuscript, "A polyvalent STING agonist prolongs innate activation against cancer". As mentioned in my previous email, having consulted with the original reviewers (whose comments you will find at the end of this message), I am pleased to say that we shall be happy to publish the manuscript in *Nature Biomedical Engineering*, provided that the points specified in the attached instructions file are addressed.

Please use the template provided to reformat your manuscript and provide all the information required. When you are ready to submit the final version of your manuscript, please upload the files specified in the instructions file.

In the meantime, we will assess the main text in more detail, in particular the title and abstract, for clarity, accessibility and readability. Please expect an update to this message with additional points to address. However, you don't need to wait for the update to act on the instructions in the attached document and to submit the final files.

For primary research originally submitted after December 1, 2019, we encourage authors to take up transparent peer review. If you are eligible and opt in to transparent peer review, we will publish, as a single supplementary file, all the reviewer comments for all the versions of the manuscript, your rebuttal letters, and the editorial decision letters. **If you opt in to transparent peer review, in the attached file please tick the box 'I wish to participate in transparent peer review'; if you prefer not to, please tick 'I do NOT wish to participate in transparent peer review'**. In the interest of confidentiality, we allow redactions to the rebuttal letters and to the reviewer comments. If you are concerned about the release of confidential data, please indicate what specific information you would like to have removed; we cannot incorporate redactions for any other reasons. If any reviewers have signed their comments to authors, or if any reviewers explicitly agree to release their name, we will include the names in the peer-review supplementary file. More information on transparent peer review is available.

Best wishes,

João

Dr João Duarte
Senior Editor, Nature Biomedical Engineering

Reviewer #1 (Report for the authors (Required)):

The authors have adequately and satisfactorily addressed the previous raised comments.

Reviewer #2 (Report for the authors (Required)):

Review of NBME-20-0300B A polyvalent STING agonist prolongs innate activation against cancer

The manuscript is a revised response to review addressing all of the key points raised by this reviewer. This reviewer is particularly impressed with the response to questions of STING agonist dose, duration of effect, and the experiments in the STING knockout mice. These clearly show that the drug acts through STING in host cells in the tumor, provides a sustained STING activation, resulting in improved mechanisms of response. These data add to the already strong manuscript, and help us interpret the data and why this is new and improved.

No significant issues.

Rebuttal 1

Jinming Gao, Ph.D.
Elaine Dewey Sammons Distinguished Chair in Cancer Research
In Honor of Eugene P. Frenkel, M.D.

Telephone: 214-645-6370
Fax: 214-645-6347
Email: jinming.gao@utsouthwestern.edu

August 6, 2020

João Duarte, Ph.D.
Senior Editor, *Nature Biomedical Engineering*

Dear João,

Thank you so much for your continued support to our manuscript “A polyvalent STING agonist prolongs innate activation against cancer”. We appreciate the helpful comments from all three reviewers and believe the revisions have significantly strengthened the manuscript. In an attached file, we provide the detailed responses to the reviewers’ comments in a point-by-point fashion. All the changes in the manuscript are highlighted in yellow.

Below we provide a summary on the main improvements in response to your inquiries:

To evaluate the dose-limiting toxicity, we performed a thorough assessment of the toxicity of all formulation groups (5% glucose as control, low/high dose of free cGAMP, PC7A polymer, cGAMP-PC7A NPs) in mouse models. We discovered that all groups except high dose of free cGAMP (50 µg) did not show any significant toxicity over controls, while the immune-related side effects were elevated at the high cGAMP dose as shown by impaired liver and kidney functions, and increased systemic cytokines (IL10) (Supplementary Fig. 10).

We clarified the different PC7A and cGAMP doses used in the study and further investigated the antitumor efficacy of cGAMP at high dose (50 µg) vs. low dose (2.5 µg) in MC38 tumor-bearing mice. We observed a marginal improvement in tumor growth inhibition but the difference is not statistically significant (Supplementary Fig. 10). Combined with toxicity data, 2.5 µg of cGAMP and 50 µg of PC7A offers the optimal base dose for cGAMP-PC7A NPs for antitumor studies, which allows for sustained activation of STING (Supplementary Fig. 9) without the introduction of systemic toxicity.

We also included a detailed description of the preparation of cGAMP-PC7A NP formulation in the “Method” section, with additional characterization (TEM, DLS and Zeta potential) in the supplementary information (Supplementary Fig. 8).

Lastly, we enlisted Dr. Yang-xin Fu, a renowned tumor immunologist at our institution, for assistance in mechanistic investigation of STING status as well as the specific immune cells that are responsible for the antitumor immunity. Data show host STING status but not that of the cancer cells is essential for the antitumor immunity. Furthermore, blocking of CD8 T cells completely abolished the antitumor immunity, and DC knockout mice showed partial inhibition. We included these data in the Supplementary Fig. 12. We also added Dr. Fu as a co-author to acknowledge his intellectual contributions in these studies.

We hope these revisions satisfactorily addressed the concerns by the reviewers. Please let me know if you need any additional information.

Sincerely,

Jinming Gao, Ph.D.

Response to Reviewers' Comments

We appreciate all three reviewers for their constructive and thoughtful comments and suggestions. We are also grateful for their appreciation of the significance of these studies on novel STING ligands and mechanisms for innate activation. We have responded to all questions, suggestions and requests for additional experiments in the revised manuscript. A detailed point-by-point response to the reviewers' feedback follows.

Reviewer #1

"This manuscript by Li et al., describes the unique role of a synthetic polymer, PC7A, to induce STING condensation through polyvalent interactions for activation of immune responses. The authors found that this PC7A polymer binds to a distinct STING surface site from the cGAMP-binding pocket, thus retaining STING activation in cGAMP-resistant clinical variants... The current work attempted to develop an in-depth understanding of how PC7A uniquely activates the STING pathway and how it can be used with cGAMP to achieve a synergistic treatment effect. Overall, this is a solid study that elucidates the mechanistic role of a previously reported PC7A polymer in acting STING. However, the innovative aspect of the current work seems incremental and falls a bit short for the intended journal. In addition, there are a number of critical issues that the authors should consider."

We appreciate the positive feedback by the reviewer. We would like to highlight several innovative aspects of the current work. First, we determined a distinctive surface binding site by the PC7A polymer that is different from cGAMP or other cyclic dinucleotides. Although previous report (*Nat. Nanotech.* 12, 648-654) showed PC7A nanoparticle vaccine worked through STING for T cell activation, it is not clear whether PC7A competes with cGAMP for the same binding site at the STING dimer interface. The non-competitive binding insight formulates a basis to combine PC7A with cGAMP for synergistic STING activation while demonstrating the ability of PC7A to activate cGAMP-resistant STING variants (Fig. 4d-f). Second, we uncovered a PC7A-induced protein condensation mechanism for STING activation. This is the first example of using a synthetic polymer to induce polyvalent phase condensation for biological activation. Phase condensates are an emerging principle that impacts a broad range of biological processes and is under intensive investigations in biophysics and cell biology (*Science* 357, 4382; *Cell* 176, 419-434). The current study provides the proof of concept to install polymer-induced protein condensation as an emerging bioengineering principle for biological interrogation and pharmaceutical development. Third, new mechanistic studies as inquired by the reviewers show the importance of host STING and CD8 T cells in cGAMP-PC7A NP therapy, further elucidating the immunological gate keepers in therapy (please see below). Together these results allowed new mechanistic insights and novel innate immunity strategies that exploit noncanonical STING activation with intrinsic pathways for cancer immunotherapy.

We clarified the novelty impact of the current work in the discussion (second paragraph on p.13).

"1. The authors claimed STING-dependent activation of immunity by PC7A, but some important results are missing to support the unique contribution of PC7A-induced STING pathways. At least, the authors should consider using STING and IFN knockout mice to confirm the importance of STING pathway."

We appreciate the suggestion by the reviewer to confirm the importance of the STING pathway in PC7A-induced antitumor immunity. We performed new studies using STING knockout mice and compared the antitumor efficacy by PC7A and cGAMP-PC7A nanoparticles over wildtype mice. Results show without treatment, colorectal MC38 cancer cells grew faster in STING knockout (*Tmem173*^{-/-}, *Tmem173* encodes STING) mice than wildtype mice, indicating the role of the STING pathway in immune protection by the host alone. Treatment by PC7A and cGAMP-PC7A nanoparticles completely lost antitumor efficacy in *Tmem173*^{-/-} animals. Moreover, PC7A-induced antitumor immunity is not dependent on the cancer cell STING status in the MC38 tumor model (please also see response to Minor Question 3 by Reviewer 2). Together, these data demonstrate that the host STING plays a critical role in the antitumor response of PC7A and cGAMP-PC7A nanoparticles.

We added the new data in the revised manuscript (fourth paragraph on p.11) and Supplementary Fig. 12a-c.

“2. Activation of STING could also promote dendritic cells, nature killer cells and cytotoxic T cells for immune response, however, none of them was examined in the current manuscript. The authors should assess the contribution of these cells in the PC7A induced antitumor efficacy.”

We appreciate the reviewer’s suggestion to investigate the immune cell-dependent mechanism of antitumor immunity. As suggested, we employed CD11c-DTR transgenic mice for depletion of dendritic cells (*Immunity* 17, 211-220), and antibody blockade of CD8 T cells and NK cells in C57/B6 mice to assess their contributions to antitumor efficacy. Blockade of CD8 T cells completely abolished the antitumor efficacy by PC7A treatment whereas blockade of NK cells showed minimal effect. CD11c-DTR mice showed that DC depletion reduced the therapeutic efficacy after treatments but to a lesser degree than the CD8 T cell blockade. Together with host STING knockout study, these results have much better defined the immunological mechanisms of PC7A beyond our previous report (*Nat. Nanotech.* 12, 648-654) showing that generation of antigen-specific T cells by the PC7A nanoparticle vaccine was STING-dependent.

We added the new data in the revised manuscript (second paragraph on p.12) and Supplementary Fig. 12d-f.

“3. Drug dosage plays a pivotal role in their therapeutic efficacy. In the current studies, the authors chose to use a much lower dose of cGAMP (2.5 µg) than PC7A (50 µg) in animal studies (see Figure 5). In Figures 5a and 6, PC7A alone works better for cGAMP alone, while in Figure 5d and 5e, cGAMP works more effectively. Some explanations are necessary. What limits the dose of cGAMP? Do different ratios of combining the two offer different treatment outcomes?”

“4. In the same vein, given that the STING agonists often show dose-limiting toxicity (as claimed by the authors in the Introduction part), the safety and immune-related side effects should be assessed.”

We had previously performed antitumor efficacy and safety studies of cGAMP at higher doses (i.e., 50 µg) in MC38 tumor-bearing mice. Results show insignificant difference in tumor growth inhibition at 50 µg versus 2.5 µg cGAMP dose after three intratumoral injections. In contrast, the toxic

side effects were elevated at the higher cGAMP dose as shown by impaired liver (ALT/AST) and kidney (urea) functions, and elevated systemic cytokines (IL10). We attribute the lack of significant antitumor efficacy and increased toxicity to the clearance of cGAMP from tumor site to systemic circulation after intratumoral injection. Based on these studies, we chose to incorporate lower dose of cGAMP in PC7A nanoparticles (i.e., 2.5 µg cGAMP in 50 µg PC7A NP) to avoid high dose cGAMP-induced immune side effects. This composition (i.e., 5% weight loading) is also compatible with conventional nano formulations with a therapeutic payload. With this cGAMP-PC7A NP formulation, results did not show any increase in immune-related toxicity over the no treatment control. We added the new data in the revised manuscript (second paragraph on p.11) and Supplementary Fig. 10.

For the seemingly “contradicting” results in Figure 5, we wish to clarify that there was no significant difference between cGAMP and PC7A groups ($P=0.53$ in Fig. 5a and 0.12 in Fig. 5d, two-tailed Student’s t-test). They showed similarly improved anti-tumor efficacy compared to the untreated control group. It is also worth noting that Figure 5a-c used the MC38 model while Figure 5d-f used the TC-1 tumor model. The distinct tumor biology of these two tumor models may account for the small difference in therapeutic effect.

“5. A minor point: survived animals should be monitored for at least several weeks after the death of last reported mice (figure 5e) to confirm no tumor recurrence.”

We appreciate the reviewer’s comment. We had monitored the survival animals for additional 3 weeks. All the survived animals were free of tumor burden. We have included the extended survival data in the revised Fig. 5b and Supplementary Fig. 11b.

“6. Figure 1 shows both cGAMP and PC7A could promote STING puncta formation rapidly, however, the expression of IFNs in cGAMP is much quicker than PC7A. Why? The authors should consider discussing their time difference in immune response.”

Due to the small size of Fig. 1a, PC7A-induced puncta formation was not clearly illustrated over cytosolic STING-GFP background. In fact, PC7A-induced puncta formed more slowly than those by cGAMP (maximal times at 24 vs. 6 hours, respectively). We attribute this time difference to two possible factors. First, endosomal escape followed by cytosolic transport to reach ER-bound STING target is likely faster for small molecule cGAMP (molecular weight 674 Da) than the PC7A polymer (21 kD). Second, cGAMP-induced conformational change of STING and subsequent oligomerization (*Nature* 567, 389-393; 567; 394-398) may also occur faster than PC7A-induced STING condensate formation for immune activation. Currently, we do not have a complete understanding of the kinetic differences of STING activation, but the inquiry spurs interests for future investigation.

Per reviewer’s suggestion, we added these discussions (third paragraph on p. 14) and increased the imaging contrast in Fig. 1a to improve clarity.

“7. The PC7A polymer has been reported previously. However, when used for a new formulation, the nanocharacteristic could vary. The authors should provide characterization results of the PC7A/cGAMP

particles (TEM, DLS, Zeta potential, etc). Also, given the high-water solubility of cGAMP, how could the authors achieve a high loading efficiency of >90% when preparing PC7A nanoparticles in aqueous solution?"

Per reviewer's suggestion, we included additional analysis (TEM, DLS and Zeta potential) of cGAMP-loaded PC7A nanoparticles in Supplementary Fig. 8. The nanoparticles are spherical with diameter at 29.9 ± 2.5 nm and zeta potential at 0.2 ± 0.2 mV. Although cGAMP is water soluble, the hydrophobic interactions between the nucleotide bases (i.e., guanine and adenine on cGAMP) and cyclic side chains of PC7A as well as balanced electrostatic interactions (i.e., between negatively charged phosphate groups on cGAMP and induced ammonium groups on PC7A) enable efficient loading.

Reviewer #2

"The manuscript describes the characterization of a novel STING-binding polymer that results in STING activation through a novel mechanism. The manuscript provides clear data showing direct interaction with STING resulting in phase shifts and activation of downstream signaling components and identifies a novel binding region. The study of STING as a therapeutic target has thus far focused on variations of the endogenous ligand, and this work represents a technological advance that has the potential to improve targeting of this pathway. There remain some issues in the approach when the drug is used as a therapy that limit the conclusions. For example, the central rationale for developing new ligands is that the current ligands exhibit dose limiting toxicities and haven't translated well to humans. This drug appears to work through identical pathways and data is not provided to suggest that this will avoid the same limitations.

Major issues.

1. The authors do not adequately explain why the combination of PC7A and the conventional ligand cGAMP are required for therapy. The manuscript suggests that 'cell intrinsic and noncanonical mechanisms' are involved, as well as increased delivery of cGAMP. However, these seem unnecessary in the in vitro studies for PC7A action. Additional studies are needed to explain whether dual STING ligand treatment is superior in vivo through some mechanism of synergy or whether simply increasing the amount of STING ligands present results in improved outcomes."

We appreciate the positive feedback from Reviewer 2 on the novel mechanism of PC7A-induced STING activation. *In vitro* studies (Fig. 1a-c) illustrate that cGAMP induced rapid STING activation within 6 h followed by quick degradation inside lysosomes after 24 h. In contrast, PC7A allowed a sustained STING activation that reached maximum at 24 h and persisted at 48 h. Combining the "burst" activation by cGAMP with the "sustained" activity by PC7A can synergize STING-mediated type I IFN response that may cover the entire period for DC maturation and T cell priming, which normally requires 1-2 days (*Nature* 427, 154-159; *Front. Immunol.* 6, 563). To evaluate the temporal response *in vivo*, we measured the interferon-stimulated genes (ifn- β and cxcl10) in both tumor and draining lymph nodes at several time points after intratumoral injection (Supplementary Fig. 9). Similar to *in vitro* studies, high dose of cGAMP treatment (50 μ g) rapidly increased ifn- β and cxcl10 expression in MC38 tumors and lymph nodes at 6 h, but this expression quickly declined at 24 and 48 h. In contrast, cGAMP-PC7A NP (at low 2.5 μ g cGAMP dose and 50 μ g PC7A) induced rapid rise of ifn- β and cxcl10 expression over PC7A (50 μ g) at 6 h, and this response was persistent over 48 h. Furthermore, there was improved antitumor efficacy by cGAMP-PC7A NPs over either cGAMP or PC7A NP alone in both MC38 and TC-1 tumors (Figure 5).

We feel that these data convincingly illustrate the synergy of combining a natural STING ligand (cGAMP) with an engineered polymer (PC7A) in antitumor immunity. Moreover, cGAMP-PC7A NPs demonstrated less systemic toxicity compared to high dose cGAMP treatment (Supplementary Fig. 10), demonstrating the advantage of less perfusion loss and reduced systemic toxicity by a nanoparticle STING agonist.

We added the new data in the results (p.9-10), discussions (first paragraph on p. 15) and Supplementary Figs. 9 and 10.

“2. In the described in vivo studies the high dose of cGAMP is 2.5µg, which is below its effective dose, so of course shows weaker effects than PC7A on cytokine release and in tumor control. Most preclinical studies use cyclic dinucleotides in the 25-100µg range, and that dropping below 10µg results in a loss of efficacy. No titration information is provided for PC7A, but uses a base dose of 50µg so it is plausible that this has very similar dose-response kinetics in vivo to the endogenous ligands. There is not enough information presented to suggest that this is an improved drug.”

We had performed high dose (50 µg) cGAMP intratumoral studies where the dose was comparable to other preclinical cyclic dinucleotide studies (*Cell Rep.* 11, 1018-1030; *Cancer Res.* 78, 6308-6319). Results showed marginal improvement in antitumor efficacy over lower dose (2.5 µg) of cGAMP in the MC38 tumor model (statistically not significant); however, the toxic side effects were significantly increased (Supplementary Fig. 10). We chose 50 µg of PC7A as a base dose for PC7A NPs and cGAMP-PC7A NPs for antitumor studies. This dose allows for sustained activation of STING without the introduction of immune-related toxicity. Also, this dose is optimal for intratumoral injections to keep the injection volume less than 50 µL per tumor.

“3. The title is only supported by the initial in vitro data on STING activation. There is no information provided to suggest that prolonged innate activation contributes to the responses against cancer, since only early peak cytokine responses are assessed. Additional experiments are necessary to determine whether PC7A exhibits a prolonged STING activation in vivo.”

Per reviewer’s suggestion, we performed additional experiments to measure the interferon-stimulated genes (ifn-β and cxcl10) in tumor and draining lymph nodes at different time points after PC7A treatment. As shown in the Supplementary Fig. 9, high dose of free cGAMP (50 µg) activated STING at early time points (6 h), while PC7A sustained STING stimulation over 48 h, similar to the *in vitro* cell culture studies (Figure 1a-c). Combining low dose of cGAMP (2.5 µg) and PC7A in the NP formulation showed synergistic STING activation to achieve both rapid and sustained STING activity (from 6 to 48 h) in both tumor and draining lymph nodes. The prolonged STING activity is beneficial for DC maturation and T cell priming, which is further supported by the long-term antitumor efficacy outcomes (Figure 5).

“4. The models do not address the issue of dose limiting toxicity, which is one of the central rationales for the development of this drug.”

We have assessed the safety and immune-related side effects, and added the new information in the manuscript (second paragraph on p.11) and Supplementary Fig. 10. Basically, low dose of cGAMP (2.5 µg), PC7A, and cGAMP-PC7A NPs did not show increased toxicity over untreated controls, while high dose of cGAMP (50 µg) showed impaired liver (ALT/AST) and kidney (urea) functions, and elevated

systemic inflammatory cytokines (IL10). Thus, through efficient cytosolic delivery of cGAMP and sustained STING activity, PC7A nanoparticles reduced the effective dose of cGAMP and systemic toxicity while maintaining robust antitumor immunity (Fig. 5).

“Minor issues

1. Prior studies have shown that cyclic dinucleotides activate IFN secretion from mouse and patient tumor explants analyzed ex vivo (PMID: 30224374). The authors should explain why cyclin dinucleotides are not effective in their experiments.”

In the prior study (PMID: 30224374), the authors used a high dose of cyclic dinucleotide (50 µg) for *ex vivo* tumor studies. In the current study, high dose of cGAMP (50 µg) did induce strong type I interferon responses in tumor and nodal tissues at 6 h time point (Supplementary Fig. 9). However, the effect went away at later time points (e.g., 24 and 48 h) likely due to drug clearance from the tumor site through blood perfusion. This is further corroborated by the elevated systemic toxicity *in vivo* (Supplementary Fig. 10).

“2. Prior studies have shown synergy between STING ligands and anti-PD1 treatment in murine models (PMID: 27821498; PMID: 28483787). The authors should cite this work in their manuscript.”

As suggested we have cited these two studies in the revised manuscript (Ref. 47 and 48).

“3. The fact that STING ligands dominantly activate non cancer cells in the tumor stroma rather than cancer cells (Figure 6) presents an undiscussed confounding issue. In particular, the mixed use of THP1 (monocytic) versus 293 and fibroblasts that are used in vitro compared to the murine cell lines that are used in vivo. These may have distinct expression of STING pathway genes and STING ligand pumps which make it difficult to extrapolate in vitro to in vivo effects. The role of cancer cell versus host STING activation would be better supported if the authors reported on STING expression by the mouse cancer cells used in the in vivo study, or used STING knockout mice or STING knockout cancer cells in vivo. However, a discussion of this issue would be sufficient.”

We appreciate the Reviewer’s inquiry on the role of STING in the host vs. cancer cells on antitumor immunity. In the response to Reviewer 1’s question 1, we showed that in STING knockout (*Tmem173*^{-/-}, *Tmem173* encodes STING) mice, PC7A or cGAMP-PC7A nanoparticle treatment did not show tumor growth inhibition compared to the untreated control. This result demonstrated that host STING played a major role in the antitumor immunity induced by the nanoparticles (Supplementary Fig. 12b). We carried out the reverse experiment using *Tmem173* knockout MC38 cancer cells inoculated in STING-wildtype mice. Results show comparable antitumor efficacy by PC7A nanoparticles in treating STING knockout MC38 tumors vs. STING wildtype MC38 tumors (Supplementary Fig. 12c).

These studies highlight the importance of host cells in PC7A-induced antitumor immunity. There are on-going efforts to identify specific cell types and subtype(s) of cells that are responsible for the STING-mediated antitumor immunity. We wish to report these studies in future publications.

We added the host STING knockout and cancer cell STING knockout data in the new Supplementary Fig. 12a-c with discussions in the revised manuscript (second paragraph on p.15).

Reviewer #3

“Li et al has shown that PC7A activate delayed STING activation compare to natural STING agonist cGAMP. STING activation though PC7A stays for more time compare to cGAMP activation. Though authors have reported a novel mechanism, but study is missing some of important controls. Authors claim that PC7A may have therapeutic implication in cGAMP nonresponsive patient. For best of my knowledge cGAMP is not being investigated in clinic. ADU-S100 is known STING and being investigated in clinic was not used in this study for comparison.”

We appreciate the Reviewer’s positive feedback. We wish to clarify that the main objective of the current study is to elucidate the biochemical mechanism of STING activation by a novel synthetic polymer over cell intrinsic pathways. We chose cGAMP as an intrinsic STING activator to compare the spatio-temporal difference of STING activation to PC7A. Although ADU-S100 has been studied in the clinic, its safety and antitumor efficacy responses are still under active investigation (i.e., high dose of ADU-S100 led to T cell apoptosis. *Cell Rep.* 25, 3074-3085). We feel that ADU-S100 warrants a separate study with careful in-depth comparisons from molecular mechanism of STING activation to safety and antitumor efficacy indications.

“Major comment:

1- One of the benefit of small molecule STING agonist is its short half-life in vivo which is prepared for safety of the patients. Authors should discuss how PC7A can be used in patients. If a low dose of PC7A therapy can improve T cell response (reported by Chandra et al. 2014 and Sivick et al 2018), please discuss it.”

We appreciate the Reviewer’s suggestion to discuss how PC7A can be used in patients. We are aware of the literature reports that low dose small molecule STING agonist is more optimal at generating antitumor immunity whereas high doses showed diminished tumor-specific T cell response (*Cancer Immunol. Res.* 2, 901-910; *Cell Rep.* 25, 3074-3085). The negative effects of high dose of STING agonist may be attributed to a STING-mediated T/myeloid cell death pathway (*Immunity* 53, 1-12; *Cell*, 171, 1110-1124; *J. Immunol. Res.* 199, 397-402). We performed the low dose PC7A treatment (5 µg) and found a reduced tumor growth inhibition compared to the 50 µg dose (see Figure below). Because of the distinctive mechanism of STING activation by the PC7A polymer, we feel that additional mechanistic studies are warranted for PC7A nanoparticles before clinical translation. For example, our preclinical studies show PC7A nanoparticles preferentially target CD45+ leucocytes with significantly increased cytokine (ifnβ and cxcl10) expression in the resected patient tumors (Figure 6e,f). Furthermore, antitumor efficacy studies show host STING and CD8 T cells played a critical role in antitumor immunity (Supplementary Fig. 12). Further characterization of immune and stromal cell types or subtypes will be helpful to establish predictable biomarkers for patient response.

Per reviewer’s suggestion, we added relevant discussions in the revised manuscript (second paragraph on p.15).

“2- 50ug of PC7A was tested to compare with 0.5ug and 2.5 ug of cGAMP or 50ug PC7A vs 80ng aGAMP. Please discuss why low dose of cGAMP were used to compare. If optimum cGAMP was titrated for in vitro and in vivo assays. Please discuss rationale for using low doses of cGAMP.”

We apologize for the confounding use of several cGAMP doses in the current study. In general, we employed low dose of cGAMP for intratumoral administration because we expect a high cytosolic concentration can be reached as aided by PC7A delivery. PC7A-induced STING activation further augments the innate stimulation that reduces the necessity of high cGAMP dose (Supplementary Fig. 9). Low cGAMP dose was chosen also because high dose cGAMP led to increased toxicity (Supplementary Fig. 10).

“Minor:

1-please mention, how many time each experiments were repeated and what is represented in figures.”

For *in vitro* studies, each experiment was repeated at least three independent times unless indicated otherwise. For *in vivo* studies, STING activity experiment was carried out in three biologically independent mice per group (Supplementary Fig. 9). For all other animal studies, each experiment was carried out in at least five independent mice per group. We added the information in the figure captions when applicable in the revised manuscript.

“2-Please also discuss, what authors think about delayed STING activation by PC7A compared to cGAMP, as cytokine was mRNA activation seen after 12 hours of treatment.”

We attribute the delayed STING activation by PC7A to two possible factors. First, endosomal escape followed by cytosolic transport to reach ER-bound STING target is likely faster for cGAMP (molecular weight 674 Da) than the PC7A polymer (21 kD). Second, cGAMP-induced conformational change of STING and subsequent oligomerization (*Nature* 567, 389-393; 567; 394-398) may also occur faster than PC7A-induced STING condensate formation for immune activation. As suggested we added relevant discussions (third paragraph on p.14) in the revised manuscript.